Regional trends in the moderate-to-vigorous intensity physical activity and screen time of Canadians before and during the COVID-19 pandemic

Saunders Travis trsaunders@upei.ca 1
Colley Rachel C. 2
1 Department of Applied Human Sciences, University of Prince Edward Island , Charlottetown , Prince Edward Island , Canada
2 Health Analysis Division, Statistics Canada , Ottawa , Ontario , Canada
Basu Arindam
Electronic publication date: 2024 Feb 29
Publication date: 2024
Volume: 12
Electronic Location ID: e16913
Received 2023 Jul 2; Accepted 2024 Jan 17
Copyright: ©2024 Saunders and Colley
Copyright year: 2024
Copyright holder: Saunders and Colley
License: This is an open access article distributed under the terms of the Creative Commons Attribution License, which permits unrestricted use, distribution, reproduction and adaptation in any medium and for any purpose provided that it is properly attributed. For attribution, the original author(s), title, publication source (PeerJ) and either DOI or URL of the article must be cited.
License URL: https://creativecommons.org/licenses/by/4.0/

Keywords: Physical activity, Sedentary Behaviour, Population Surveillance, COVID-19, Canada, Screen time

Funding: The authors received no funding for this work.

==============================
Background

During the COVID-19 pandemic, public health approaches and disease-transmission varied widely across Canadian regions. This may have led to different trajectories for moderate-to-vigorous intensity physical activity (MVPA) and screen time during this period. The purpose of this investigation was to describe age- and gender-specific regional trends in MVPA and screen time for Canadian youth (ages 12–17 years) and adults (ages 18+) from 2018 to 2021.

Methods

Data was collected using the Canadian Community Health Survey, which includes representative data for 5 distinct regions: Atlantic Canada, Québec, Ontario, the Prairie Provinces, and British Columbia (BC). Participants aged 12+ in each region self-reported their total daily screen time, as well as MVPA in 5 domains: overall, recreational, school, occupational/household and active transportation. Results were compared for 2018 (pre-pandemic), January–March of 2020, September–December of 2020, and 2021 using repeated measures t-tests.

Results

Among youth, all regions except for Atlantic Canada and BC experienced significant reductions in the proportion of youth meeting MVPA recommendations in the fall of 2020 (all p < 0.001), although these had returned to baseline for all regions except Ontario by 2021. Trends varied across regions among adults aged 18–64 years. In Québec, there was 7-percentage point reduction in the proportion of males meeting the MVPA recommendations in the fall of 2020 compared to 2018, while there was a 4-percentage point increase among females in 2021 (all p < 0.05). In Ontario and the Prairie provinces, males saw a 4-percentage point decrease in activity recommendation adherence in 2021, when compared to 2018 (p < 005). There were no other significant differences for any region when comparing the fall of 2020 or 2021 with 2018 (all p > 0.05). Among adults aged 65+ years, significant increases in MVPA were observed in Atlantic Canada and the Prairies in the fall of 2020, and in Atlantic Canada, Québec and the Prairies in 2021 (all p < 0.05). With limited exceptions, self-reported screen time increased significantly across regions and age groups for both males and females (all p < 0.05).

Conclusions

MVPA levels of Canadians during the COVID-19 pandemic varied both by region and age group. Self-reported MVPA of Canadian youth dropped in most regions in the fall of 2020, before returning to pre-pandemic levels in 2021. Activity levels of Canadians aged 18–64 years were relatively stable during the pandemic and increased for Canadians aged 65+ in most regions. Differences in trajectories across genders observed at the national level were often less apparent in individual regions. Recreational screen use increased across all regions, ages and genders with very few exceptions. These results highlight the differences and similarities in activity and screen time trajectories across the Canadian population and suggest the need for additional research to identify best practices for promoting healthy movement behaviours during future pandemics.

Introduction

Low levels of moderate-to-vigorous intensity physical activity (MVPA) and high levels of recreational screen time are associated with increased risk of chronic disease morbidity and mortality (Warburton, Nicol & Bredin, 2006; Carson et al., 2016; Ekelund et al., 2016; Saunders et al., 2020). To maximize health, the Canadian 24-Hour Movement Guidelines recommend that youth aged 12–17 years accumulate at least 60 min per day of MVPA, and no more than 2 h of recreational screen time (Tremblay et al., 2016). Adults aged 18+ should accumulate at least 150 min of moderate-to-vigorous activity per week, along with no more than 3 h per day of recreational screen time (Ross et al., 2020). Data collected early in the COVID-19 pandemic suggested rapid and significant reductions in the proportion of Canadians meeting recommendations for MVPA and screen time among Canadians of all ages (Rhodes et al., 2020; Moore et al., 2020; Moore et al., 2021; Caldwell et al., 2022; Colley & Saunders, 2023a; Colley & Saunders, 2023b; Liu et al., 2023). Recent investigations using the Canadian Community Health Survey (CCHS) have suggested that the MVPA levels of Canadian adults and male youth returned to or surpassed pre-pandemic levels by 2021, although the activity levels of female youth remained depressed (Colley & Saunders, 2023a; Colley & Saunders, 2023b). When examined at the national level, recreational screen time remained significantly higher in 2021 than in 2018 for all Canadian age groups (Colley & Saunders, 2023a; Colley & Saunders, 2023b). The above studies have identified nation-wide trends in MVPA and recreational screen time but did not examine regional trends. To our knowledge only one study has examined regional differences in MVPA and screen time among Canadian children and youth during the COVID pandemic (Caldwell et al., 2022), and reported that individuals in the Prairies and Atlantic Canada accumulated more MVPA than those living in Québec. However, to date no study has examined regional differences among adults, nor compared regional trends using data collected in real time for any age group. This is important, as policy approaches, and the spread of virus itself, varied greatly across regions throughout the pandemic (Cameron-Blake et al., 2021; Bignami, 2021).

Canada is a decentralized federation, with the provision of healthcare falling under the responsibility of the provinces and territories, while public health is a shared responsibility between municipal, provincial and federal governments (Canada, 2011). Canadian provinces also differ greatly in terms of demographics and population density, all contributing to different approaches and experiences across regions. In July of 2020 the 4 (largely rural) Atlantic provinces formed the “Atlantic Bubble” and took a “COVID-zero” approach throughout much of 2020 and 2021. This approach dramatically limited travel into the region and saw targeted restrictions and/or closures in response to signs of community transmission. With the exception of the above targeted and time-limited restrictions, schools, child-care and recreation facilities largely remained open in Atlantic Canada from the fall of 2020 onwards (Canadian Institute for Health Information, 2023). The combination of relatively few restrictions within the region, as well as limited community spread compared to other Canadian regions, led to a comparatively higher “normalcy” in Atlantic Canada than other Canadian regions for much of 2020 and 2021 (Cameron-Blake et al., 2021). In contrast, the province of Quebec, which has large population centers in both Montreal and Quebec City, was among the hardest hit regions of North America early in the COVID-19 pandemic, leading to both widespread restrictions and high levels of community spread throughout 2020 and 2021 (Cameron-Blake et al., 2021; Canadian Institute for Health Information, 2023). Ontario, Canada’s largest province, and the three relatively rural Prairie Provinces, also experienced relatively higher community transmission throughout 2020 and 2021, while enacting less stringent restrictions than in Quebec or Atlantic Canada early in the pandemic (Cameron-Blake et al., 2021). In particular, Alberta’s approach focused on supporting businesses and encouraging personal responsibility (Cameron-Blake et al., 2021). Finally, the policy approaches in BC was a roughly similar level of stringency as Ontario and the Prairie provinces, despite experiencing relatively lower community spread throughout 2020 and 2021 (Cameron-Blake et al., 2021). As noted by Cameron-Blake et al. (2021), in addition to differing in terms of policy approaches to COVID-19, provinces also varied in terms of COVID messaging. In particular, they highlight the examples of BC and the Atlantic province of Prince Edward Island, which had relatively low transmission throughout the pandemic, and where messaging was led by public health officials. In contrast, in Quebec, Ontario and Alberta, messaging was often led by the provincial premiers, which may have led to greater confusion (Cameron-Blake et al., 2021).

It is likely that the above regional differences in policy and COVID-19 transmission may have led to different trajectories in the MVPA and screen time of individuals living in each Canadian regions. Therefore, the purpose of this article was to describe age- and gender-specific regional trends in MVPA and screen time for Canadians aged 12+ years from 2018 to 2021.

Materials & Methods

Data source

The CCHS is an annual cross-sectional survey that collects information related to health status, health care utilization, and health determinants for the Canadian population (Statistics Canada, 2019). The CCHS is collected under the Statistics Act, and therefore does not require Research Ethics Board approval. All participants provided written informed consent prior to data collection. The present analysis includes self-reported MVPA and screen time data for Canadians aged 12+ years. The primary comparison of the present analysis is between the full annual dataset collected in 2018 (pre-pandemic, n = 54,045) and 2021 (during pandemic, n = 49,243). Additional comparisons are made to sub-annual datasets collected in 2020: pre-pandemic (January to March 2020, n = 14,844) and during pandemic (September to December 2020, n = 27,234). The response rate was as follows: 58.8% in 2018, 45.6% in January to March 2020, 27.4% in September to December 2020, and 28.4% in January 2021 to February 2022. The COVID-19 pandemic had major impacts on the data collection operations for CCHS 2020. Important analytical and data quality implications related to the 2020 data are described elsewhere (Statistics Canada, 2019; Colley & Watt, 2022). Briefly, CCHS data collection was paused from March to August 2020, and this resulted in a small pre-pandemic dataset spanning January to March 2020. The CCHS data collection re-started in September 2020 and continued until December 2020. Screen time data were not collected in 2020 in a full sample and therefore are not included in the present analysis.

MVPA and screen time questions

CCHS respondents were asked to provide estimates of time (hours and minutes) spent in the past 7 days engaged in transportation, recreational, occupational/household and school-based MVPA (Statistics Canada, 2019). The specific wording of individual questions are provided in Table 1. Values greater than 2 h per day of any domain were flagged as outliers and re-coded to 2 h. This occurred in less than 4% of respondents. Youth were classified as meeting the MVPA recommendation if their average daily quantity of MVPA (including all domains of MVPA) was equal to or greater than 60 min (Tremblay et al., 2016). Adults were classified as meeting the MVPA recommendation if their weekly sum of MVPA (including all domains of MVPA) was equal to or greater than 150 min (Ross et al., 2020).

Table 1 Questions used to assess physical activity and sedentary behaviour in the CCHS (Government of Canada, 2019).

Domain of physical activity & sedentary behaviour	Question	
Youth aged 12–17 years		
Pre-Amble	The following questions are about various types of physical activities that you have done each day in the past week.	
Transportation	In the last 7 days, did you use active ways like walking or cycling to get to places such [school,] the bus stop, the shopping centre, [work] or to visit friends?
[If yes] How much time did you spend using active ways to get to places...	
School-based	In the last 7 days, did you do sports, fitness or recreational physical activities while at [school or day camp], including during physical education classes, during your breaks and any other time you played indoors or outdoors / day camp, including any time you played indoors or outdoors]?
[If yes] Did any of these activities make you sweat at least a little and breathe harder?
[If yes] How much time did you spend doing these activities at [school / day camp/school or day camp] that made you sweat at least a little and breathe harder...	
Recreation	In the last 7 days, did you do physical activities in your leisure time including exercising, playing an organized or non-organized sport or playing with your friends? [If yes] Did any of these leisure-time activities make you sweat at least a little and breathe harder?
How much time did you spend doing these leisure-time activities that made you sweat at least a little and breathe harder...	
Household/Occupational	In the last 7 days, did you do any other physical activities [that you have not already reported], for example, while you were [doing paid or unpaid work or] helping your family with chores?
[If yes] Did any of these other physical activities make you sweat at least a little and breathe harder?
[If yes] How much time did you spend doing these other physical activities that made you sweat at least a little and breathe harder...	
Sedentary Behaviour	On a [school or work day / day that was not a school or workday], how much of your free time did you spend watching television or a screen on any electronic device while sitting or lying down?	
Adults aged 18+ years		
Pre-Amble	The following questions are about various types of physical activities done in the last 7 days. I want you to only think of activities you did for a minimum of 10 continuous minutes.	
Transportation	In the last 7 days, that is from last [Day of the week 7 days ago] to yesterday, did you use active ways like walking or cycling to get to places such as work, school, the bus stop, the shopping centre or to visit friends?
[If yes] How much time in total, in the last 7 days, did you spend doing these activities? Please only include activities that lasted a minimum of 10 continuous minutes.	
Recreation	[Not including activities you just reported,] in the last 7 days, did you do sports, fitness or recreational physical activities, organized or non-organized, that lasted a minimum of 10 continuous minutes? Examples are walking, home or gym exercise, swimming, cycling, running, skiing, dancing and all team sports.
[If yes] Did any of these recreational physical activities make you sweat at least a little and breathe harder?
[If yes] In the last 7 days, how much time in total did you spend doing these activities that made you sweat at least a little and breathe harder?	
Occupational/Household	In the last 7 days, did you do any other physical activities while at work, in or around your home or while volunteering? Examples are carrying heavy loads, shoveling, and household chores such as vacuuming or washing windows. Please remember to only include activities that lasted a minimum of 10 continuous minutes.
[If yes] Did any of these other physical activities make you sweat at least a little and breathe harder?
[If yes] In the last 7 days, how much time in total did you spend doing these activities that made you sweat at least a little and breathe harder?	
Sedentary Behaviour	On a [school or work day / day that was not a school or workday], how much of your free time did you spend watching television or a screen on any electronic device while sitting or lying down?	

Youth aged 12–17 years were asked to estimate their average daily recreational screen time for days they went to school and days they did not go to school (≤2 h per day, 2 to ≤4, 4 to ≤6, 6 to ≤8 and 8+ hours per day). Adults aged 18+ were asked to estimate their average daily recreational screen time for days that they went to work and days they did not go to work using the same categories. Screen time categories were re-coded to: ≤2 h per day, 2 to ≤4, 4+ hours per day for all respondents. Youth averaging ≤2 h per day were classified as meeting screen time recommendations for their age group (Tremblay et al., 2016). None of the screen time categories used by the CCHS align with the screen time recommendation for adults within the Canadian 24-Hour Movement Guidelines (≤3 h per day Ross et al., 2020), therefore adherence to this benchmark was not assessed in the present analysis.

Statistical analysis

Descriptive statistics were used to produce weighted means of hours of screen time and weighted percentages of those accumulating ≤2 h per day, 2 to ≤4, 4+ hours per day for all respondents. Results are presented as youth 12–17 years, adults 18–64 years, and adults 65+ years. The CCHS gathers health-related data at sub-provincial levels of geography to provide estimates representative of health regions or combined health regions. The present analysis compares five regions of Canada moving from east to west: Atlantic Canada (Newfoundland & Labrador, Nova Scotia, New Brunswick, and Prince Edward Island), Québec, Ontario, the Prairie Provinces (Saskatchewan, Manitoba and Alberta), and British Columbia (BC). The Canadian Territories of the Yukon, Northwest Territories and Nunavut were excluded from this analysis because two years of data collection are required to produce reliable estimates in the territories. Excluded from the survey’s coverage are: person living on reserves and other Aboriginal settlements in the provinces; full-time members of the Canadian Forces; the institutionalized population, and persons living in the Québec health regions of Région du Nunavik and Région des Terres-Cries-de-la-baie-James. Altogther, these exclusions represent less than 3% of the Canadian population aged 18 and over (Statistics Canada, 2019).

Variance of the estimates was examined using 95% confidence intervals with bootstrap weights applied. Survey weights were applied to the data to address non-response bias and to make the results representative of the Canadian population living in the ten provinces, as described in detail elsewhere (Statistics Canada, 2019). Analyses were conducted using SAS (Version 9.4) and differences between the three time periods were tested using contrast statements within the PROC DESCRIPT procedure in SAS-callable SUDAAN (version 11.0.3).

Results

Participant demographics

The mean age of participants across all time points ranged between 14.4–14.7 years for youth, 41.2–41.5 years for adults 18–64 years, and 73.7–74.2 for adults 65+. Across all ages and time-points, the sample ranged from 48.6–53.6% female, and was 79.0–85.8% urban. The highest level of household education varied by age group: among youth 83.6–86.9% of respondents came from a household with at least some post-secondary education across all time points. Among adults aged 18–64 years, this varied from 67.6–71.2%, and among adults 65+ years it varied from 53.6%–57.5%.

Overall MVPA

National and regional trends in MVPA are presented in Tables 2 and 3 and Fig. 1. At the national level, the proportion of youth meeting Canada’s MVPA recommendation dropped by 14.8 and 9.9 percentage points among boys and girls respectively between 2018 and fall of 2020 at the national level (all p < 0.001), and returned to pre-pandemic levels by the fall of 2021 for boys, but not girls. At the regional level, Ontario and the Prairie Provinces saw significant reductions in the proportion of youth meeting the recommendations in late 2020 in both genders, while a decrease was seen for boys only in Québec (all p < 0.001). By 2021 the number of youth meeting MVPA guidelines had returned to pre-pandemic levels for boys and girls in all provinces outside of Ontario. In BC and Atlantic Canada, there were no significant changes in MVPA levels when comparing either the fall of 2020 or 2021 with the pre-pandemic period (2018) in either gender (all p > 0.05).

Table 2 Proportion of participants meeting Canada’s physical activity recommendations before and during the COVID-19 pandemic.

	2018	Jan–Mar 2020	Sept–Dec 2020	2021	
	n	Estimate	n	Estimate	n	Estimate	n	Estimate	
12–17 years									
Canada-wide	3,952	49.6 (47.3, 51.9)	911	53.7 (49.5, 58.0)	1,573	37.2 (34.2, 40.3)*	3,501	43.8 (41.2, 46.4)*	
Males	2,024	54.3 (51.2, 57.4)	465	60.0 (53.9, 65.8)	813	39.5 (35.5, 43.8)*	1,809	52.2 (48.6, 55.9)	
Females	1,928	44.7 (41.3, 48.1)	446	47.1 (41.3, 53.1)	760	34.8 (30.6, 39.2)*	1,692	35.0 (31.7, 38.4)*	
Atlantic Canada	496	44.0 (38.9, 49.2)	123	46.9 (37.3, 56.7)	170	42.1 (33.5, 51.3)	422	45.8 (40.1, 51.6)	
Males	258	46.1 (39.2, 53.1)	62	52.5 (38.4, 66.2)	81	39.0 (26.8, 52.7)	217	49.4 (41.3, 57.6)	
Females	238	41.7 (34.5, 49.2)	61	41.0 (28.3, 55.1)	89	45.0 (33.6, 57.0)	205	41.9 (34.1, 50.2)	
Québec	843	45.0 (40.3, 49.7)	206	48.6 (40.9, 56.4)	386	29.8 (24.7, 35.5)*	792	43.4 (38.7, 48.3)	
Males	417	52.0 (45.6, 58.3)	107	48.9 (38.4, 59.4)	190	31.3 (24.2, 39.4)*	393	51.2 (43.8, 58.6)	
Females	426	37.6 (31.3, 44.4)	99	48.4 (36.8, 60.1)	196	28.3 (21.3, 36.5)	399	35.4 (29.4, 41.8)	
Ontario	1,185	51.7 (47.3, 56.0)	258	53.5 (45.2, 61.6)	413	36.9 (31.1 43.0)*	1,003	37.9 (33.3, 42.7)*	
Males	618	56.3 (50.4, 62.0)	137	63.1 (50.7, 74.0)	212	37.8 (29.9, 46.4)*	526	46.8 (40.2, 53.5)*	
Females	567	46.8 (40.6, 53.0)	121	43.4 (32.2, 55.2)	201	35.9 (28.0, 44.6)*	477	28.7 (23.0, 35.1)*	
Prairie Provinces	896	51.3 (46.9, 55.6)	196	54.5 (46.5, 62.2)	446	36.2 (31.2, 41.4)*	767	49.5 (44.6, 54.4)	
Males	458	54.0 (48.0, 59.9)	92	60.6 (49.0, 71.2)	245	39.3 (32.6, 46.5)*	391	56.3 (49.4, 63.0)	
Females	438	48.4 (42.2, 54.6)	104	47.9 (37.4, 58.6)	201	32.8 (25.7, 40.9)*	376	42.5 (35.7, 49.6)	
British Columbia	532	51.2 (45.5, 56.8)	128	65.6 (56.6, 73.5)*	158	50.4 (42.3, 58.4)	517	52.8 (47.5, 58.1)	
Males	273	56.6 (48.2, 64.6)	67	71.7 (59.0, 81.8)*	85	59.6 (48.4, 69.9)	282	66.3 (58.0, 73.7)	
Females	259	45.4 (37.7, 53.3)	61	59.0 (45.6, 71.2)	73	40.5 (29.5, 52.6)	235	38.8 (31.5, 46.8)	
18–64 years									
Canada-wide	34,040	58.7 (57.7, 59.6)	7,468	56.5 (54.6, 58.4)	11,459	57.0 (55.5, 58.5)	26,797	57.5 (56.5, 58.4)	
Males	15,898	63.0 (61.6, 64.3)	3,485	60.2 (57.5, 62.8)	5,191	59.2 (56.9, 61.3)*	12,199	59.8 (58.4, 61.3)*	
Females	18,142	54.3 (53.1, 55.6)	3,983	52.9 (50.2, 55.5)	6,268	54.9 (52.8, 56.9)	14,598	55.1 (53.6, 56.5)	
Atlantic Canada	4,122	56.5 (54.4, 58.7)	792	62.4 (56.6, 67.9)	1,119	56.3 (52.1, 60.3)	5,346	58.4 (56.5, 60.2)	
Males	1,800	59.0 (55.9, 61.9)	339	72.2 (63.4, 79.5)*	510	53.7 (47.1, 60.2)	2,360	62.0 (59.1, 64.8)	
Females	2,322	54.2 (51.3, 57.1)	453	52.9 (45.8. 59.9)	609	58.7 (53.4, 63.8)	2,986	54.9 (52.4, 57.3)	
Québec	7,595	55.6 (53.8, 57.3)	1,609	59.1 (55.3, 62.9)	2,556	52.5 (49.1, 55.8)	3,914	56.2 (53.9, 58.4)	
Males	3,642	61.2 (58.8, 63.6)	751	65.3 (60.0, 70.3)	1,192	54.6 (50.1, 59.0)*	1,839	58.3 (54.9, 61.5)	
Females	3,953	49.8 (47.4, 52.3)	585	52.8 (47.9, 57.7)	1,364	50.3 (45.6, 54.9)	2,075	54.1 (51.0, 57.1)*	
Ontario	9,875	57.2 (55.5, 58.8)	2,514	53.4 (50.2, 56.6)*	3,927	54.6 (52.0, 57.2)	7,182	54.8 (53.0, 56.6)	
Males	4,548	61.4 (59.0, 63.8)	1,186	55.3 (50.5, 59.9)*	1,765	57.9 (53.9, 61.8)	3,216	57.0 (54.5, 59.6)*	
Females	5,327	53.0 (50.6, 55.3)	1,328	51.6 (47.0, 56.1)	2,162	51.4 (47.9, 54.9)	3,966	52.6 (50.0, 55.3)	
Prairie Provinces	7,742	59.7 (58.0, 61.3)	1,592	54.0 (49.6, 58.3)*	2,474	59.3 (56.5, 62.1)	7,731	58.4 (56.6, 60.2)	
Males	3,705	64.3 (61.8, 66.7)	757	57.5 (51.5, 63.2)*	1,134	61.8 (57.5, 65.8)	3,582	60.3 (57.8, 62.8)*	
Females	4,037	54.9 (52.6, 57.2)	835	50.4 (45.2, 55.7)	1,340	56.8 (53.1, 60.4)	4,149	56.4 (54.0, 58.8)	
British Columbia	4,706	68.0 (65.7, 70.2)	961	62.2 (57.3, 66.8)*	1,383	69.0 (65.0, 72.8)	2,624	65.5 (62.7, 68.1)	
Males	2,203	70.7 (67.4, 73.8)	452	64.2 (56.9, 70.9)	590	69.7 (63.2, 75.6)	1,202	69.0 (65.2, 72.5)	
Females	2,503	65.3 (62.4, 68.2)	509	60.2 (53.7, 66.4)	792	68.3 (63.2, 73.0)	1,422	62.0 (58.3, 65.6)	
65+ years									
Canada-wide	16,053	36.8 (35.6, 38.0)	6,465	38.9 (36.8, 41.0)	14,202	39.9 (38.5, 41.4)*	18,945	40.1 (38.9, 41.3)*	
Males	7,067	40.7 (38.9, 42.5)	2,821	43.8 (40.6, 47.1)	6,063	44.9 (42.7, 47.0)*	8,254	43.8 (42.1, 45.6)*	
Females	8,986	33.4 (31.8, 35.0)	3,644	34.6 (32.0, 37.2)	8,139	35.6 (33.7, 37.6)	10,691	36.9 (35.3, 38.5)*	
Atlantic Canada	2,259	31.5 (29.0, 34.0)	1,859	34.8 (32.1, 37.5)	4,509	36.9 (34.8, 39.0)*	4,113	35.4 (33.4, 37.4)*	
Males	1,002	37.6 (33.8, 41.6)	824	38.1 (33.8, 42.6)	1,933	40.8 (37.7, 44.0)	1,788	41.9 (39.0, 45.0)	
Females	1,257	26.1 (23.2, 29.2)	1,035	31.8 (28.3, 35.6)*	2,576	33.4 (30.5, 36.4)*	2,325	29.6 (27.1, 32.2)	
Québec	3,582	32.3 (30.0, 34.6)	907	36.6 (32.8, 40.6)	1,762	33.3 (30.1, 36.5)	2,641	36.4 (34.0, 38.9)*	
Males	1,613	37.6 (33.9, 41.3)	419	38.1 (31.5, 45.1)	779	37.9 (33.0, 43.1)	1,156	42.1 (38.2, 46.0)	
Females	1,969	27.7 (25.1, 30.6)	488	35.4 (30.0, 41.1)*	983	29.2 (25.2, 33.5)	1,485	31.4 (28.3, 34.7)	
Ontario	4,890	35.9 (33.6, 38.3)	1,357	37.7 (33.7, 41.9)	2,966	38.9 (36.1, 41.8)	5,430	37.8 (35.8, 39.8)	
Males	2,098	38.3 (35.0, 41.8)	591	46.4 (39.8, 53.2)*	1,265	45.1 (40.9, 49.3)*	2,347	39.7 (36.7, 42.8)	
Females	2,792	33.9 (30.8, 37.2)	766	30.3 (25.7, 35.3)	1,701	33.7 (30.1, 37.4)	3,083	36.2 (33.5, 39.0)	
Prairie Provinces	3,064	36.4 (33.8, 39.0)	1,715	38.1 (34.9, 41.3)	3,413	42.6 (40.0, 45.3)*	4,669	44.3 (42.0, 46.7)*	
Males	1,341	42.3 (38.4, 46.3)	706	47.3 (41.9, 52.7)	1,428	48.0 (44.1, 51.9)*	2,040	49.1 (45.8, 52.5)*	
Females	1,723	31.2 (27.9, 34.6)	1,009	29.9 (26.2, 33.9)	1,985	37.9 (34.5, 41.4)*	2,629	40.1 (36.8, 43.4)*	
British Columbia	2,258	50.4 (47.4, 53.5)	627	49.1 (43.9, 54.2)	1,552	53.3 (49.7, 56.9)	2,092	50.7 (47.3, 54.1)	
Males	1,013	52.2 (47.9, 56.6)	281	46.3 (39.0, 53.9)	658	55.1 (49.4, 60.6)	923	53.1 (47.7, 58.4)	
Females	1,245	48.8 (44.8, 52.8)	346	51.5 (44.1, 58.9)	894	51.6 (46.8, 56.4)	1,169	48.6 (44.4, 52.8)	
Notes.

Data are presented as mean (95% CI).

* Significantly different from 2018, p <0.05.

Table 3 Minutes per day spent engaging in overall physical activity before and during the COVID-19 pandemic.

	2018	Jan–Mar 2020	Sept–Dec 2020	2021	
	n	Estimate	n	Estimate	n	Estimate	n	Estimate	
12–17 years									
Canada-wide	3,952	72.3 (69.5, 75.1)	911	75.1 (70.2, 80.0)	1,573	56.3 (52.6, 60.0)*	3,501	67.1 (63.8, 70.4)*	
Males	2,024	80.3 (76.0, 84.6)	465	81.8 (74.4, 89.1)	813	61.0 (55.4, 66.5)*	1,809	78.3 (73.0, 83.6)	
Females	1,928	63.8 (60.1, 67.6)	446	68.1 (61.6, 74.7)	760	51.5 (46.5, 56.4)*	1,692	55.5 (51.8, 59.2)*	
Atlantic Canada	496	63.2 (56.6, 69.7)	123	68.2 (56.7, 79.8)	170	58.0 (48.9, 67.2)	422	69.7 (62.3, 77.0)	
Males	258	68.4 (58.5, 78.2)	62	76.4 (59.7, 93.1)	81	63.2 (48.0, 78.3)	217	76.0 (64.9, 87.0)	
Females	238	57.5 (49.4, 65.6)	61	59.7 (43.7, 75.6)	89	53.3 (42.5, 64.1)	205	62.9 (53.0, 72.9)	
Québec	843	64.2 (58.9, 69.5)	206	70.7 (61.7, 79.7)	386	49.1 (43.3, 54.9)*	792	65.1 (57.0, 73.1)	
Males	417	73.4 (65.9, 80.8)	107	69.5 (56.6, 82.3)	190	50.3 (41.0, 59.6)*	393	76.8 (62.3, 91.3)	
Females	426	54.6 (47.1, 62.1)	99	71.9 (59.2, 84.7)*	196	47.9 (40.7, 55.1)	399	52.9 (46.5, 59.3)	
Ontario	1,185	74.2 (68.7, 79.8)	258	72.3 (62.8, 81.9)	413	55.3 (47.9, 62.8)*	1,003	57.8 (52.6, 62.9)*	
Males	618	81.7 (73.4, 90.0)	137	85.5 (71.2, 99.8)	212	57.6 (46.4, 68.9)*	526	68.3 (60.9, 75.8)*	
Females	567	66.4 (59.5, 73.2)	121	58.5 (46.3, 70.7)	201	53.0 (42.6, 63.4)*	477	46.8 (40.2, 53.4)*	
Prairie Provinces	896	76.7 (71.0, 82.3)	196	76.1 (66.6, 85.6)	446	55.0 (48.7, 61.4)*	767	80.1 (73.0, 87.2)	
Males	458	85.2 (76.3, 94.1)	92	78.2 (64.6, 91.7)	245	60.5 (51.1, 69.9)*	391	90.7 (79.3, 102.0)	
Females	438	67.6 (60.7, 74.5)	104	73.9 (61.3, 86.5)	201	49.3 (40.6, 58.0)*	376	69.1 (60.5, 77.7)	
British Columbia	532	77.4 (70.1, 84.6)	128	93.5 (80.4, 106.6)*	158	73.1 (63.5, 82.7)	517	78.3 (71.4, 85.1)	
Males	273	85.8 (74.5, 97.1)	67	98.9 (81.1, 116.6)	85	89.5 (75.2, 103.8)	282	93.7 (83.2, 104.1)	
Females	259	68.5 (58.7, 78.3)	61	87.9 (68.7, 107.2)	73	55.6 (43.6, 67.6)	235	62.3 (53.2, 71.4)	
18–64 years									
Canada-wide	34,040	45.2 (44.3, 46.1)	7,468	41.6 (39.9, 43.3)*	11,459	44.0 (42.7, 45.4)	26,797	44.6 (43.5, 45.6)	
Males	15,898	51.9 (50.5, 53.3)	3,485	47.6 (45.1, 50.1)*	5,191	49.8 (47.5, 52.1)	12,199	49.4 (47.8, 51.1)*	
Females	18,142	38.5 (37.4, 39.5)	3,983	35.6 (33.4, 37.7)*	6,268	38.3 (36.6, 40.0)	14,598	39.7 (38.3, 41.0)	
Atlantic Canada	4,122	45.8 (43.6, 48.0)	792	49.7 (43.8, 55.7)	1,119	44.9 (41.0, 48.7)	5,346	45.8 (43.9, 47.8)	
Males	1,800	51.0 (47.7, 54.3)	339	61.4 (52.5, 70.3)*	510	48.5 (42.1, 55.0)	2,360	52.3 (49.0, 55.7)	
Females	2,322	40.8 (38.1, 43.4)	453	38.3 (31.1, 45.6)	609	41.3 (37.1, 45.6)	2,986	39.5 (37.4, 41.6)	
Québec	7,595	39.3 (37.8, 40.8)	1,609	39.8 (37.0, 42.7)	2,556	37.7 (34.9, 40.5)	3,914	41.1 (38.9, 43.2)	
Males	3,642	46.2 (43.9, 48.6)	751	48.8 (44.1, 53.6)	1,192	43.7 (39.1, 48.2)	1,839	46.3 (42.8, 49.7)	
Females	3,953	32.2 (30.3, 34.0)	585	30.7 (27.7, 33.7)	1,364	31.6 (28.2, 35.0)	2,075	35.7 (33.4, 38.1)*	
Ontario	9,875	44.6 (43.0, 46.3)	2,514	37.5 (34.8, 40.2)*	3,927	42.3 (39.8, 44.8)	7,182	41.7 (39.9, 43.6)*	
Males	4,548	51.8 (49.2, 54.4)	1,186	41.8 (37.4, 46.2)*	1,765	48.6 (44.5, 52.7)	3,216	44.8 (42.1, 47.5)*	
Females	5,327	37.6 (35.8, 39.4)	1,328	33.2 (29.9, 36.5)*	2,162	36.1 (33.1, 39.1)	3,966	38.7 (36.0, 41.3)	
Prairie Provinces	7,742	47.2 (45.6, 48.9)	1,592	43.4 (39.4, 47.3)	2,474	45.3 (42.6, 48.0)	7,731	47.2 (45.3, 49.1)	
Males	3,705	54.0 (51.3, 56.7)	757	49.6 (44.1, 55.1)	1,134	50.6 (46.3, 55.0)	3,582	52.9 (50.0, 55.8)	
Females	4,037	40.2 (38.1, 42.4)	835	36.9 (32.1, 41.7)	1,340	39.8 (36.6, 42.9)	4,149	41.4 (39.2, 43.7)	
British Columbia	4,706	53.8 (51.5, 56.1)	961	50.6 (46.1, 55.2)	1,383	57.9 (53.5, 62.3)	2,624	54.2 (51.1, 57.3)	
Males	2,203	59.5 (56.2, 62.8)	452	53.7 (47.0, 60.4)	590	63.2 (56.4, 69.9)	1,202	62.1 (57.2, 67.1)	
Females	2,503	48.2 (45.2, 51.1)	509	47.6 (41.3, 54.0)	792	52.7 (47.3, 58.0)	1,422	46.4 (42.7, 50.1)	
65+ years									
Canada-wide	16,053	26.4 (25.3, 27.5)	6,465	26.2 (24.6, 27.7)	14,202	30.2 (28.8, 31.5)*	18,945	31.1 (30.0, 32.1)*	
Males	7,067	30.7 (28.8, 32.5)	2,821	30.3 (27.9, 32.7)	6,063	35.2 (33.0, 37.4)*	8,254	35.2 (33.6, 36.9)*	
Females	8,986	22.7 (21.5, 24.0)	3,644	22.6 (20.7, 24.5)	8,139	25.8 (24.1, 27.4)*	10,691	27.5 (26.0 28.9)*	
Atlantic	2,259	24.2 (22.1, 26.3)	1,859	24.4 (22.2, 26.6)	4,509	28.3 (26.3, 30.4)*	4,113	27.8 (25.7, 29.9)*	
Males	1,002	29.9 (26.5, 33.3)	824	28.1 (24.4, 31.8)	1,933	34.3 (31.1, 37.5)	1,788	35.4 (31.8, 39.0*	
Females	1,257	19.2 (16.7, 21.6)	1,035	21.2 (18.4, 24.0)	2,576	23.1 (20.7, 25.5)*	2,325	21.1 (19.0, 23.1)	
Québec	3,582	21.7 (20.0, 23.3)	907	23.7 (21.0, 26.4)	1,762	22.0 (19.8, 24.2)	2,641	26.8 (24.9, 28.8)*	
Males	1,613	26.4 (23.7, 29.2)	419	24.9 (20.7, 29.2)	779	25.0 (21.4, 28.6)	1,156	31.0 (27.9, 34.1)*	
Females	1,969	17.6 (15.8, 19.4)	488	22.6 (18.4, 26.9)*	983	19.4 (16.7, 22.2)	1,485	23.0 (20.6, 25.5)*	
Ontario	4,890	26.4 (24.1, 28.7)	1,357	24.8 (21.9, 27.7)	2,966	30.7 (28.0, 33.5)*	5,430	28.3 (26.5, 30.1)	
Males	2,098	29.9 (26.2, 33.7)	591	31.6 (26.8, 36.5)	1,265	37.8 (33.2, 42.4)*	2,347	30.7 (27.9, 33.4)	
Females	2,792	23.4 (20.9, 25.9)	766	18.9 (15.6, 22.3)*	1,701	24.7 (21.6, 27.9)	3,083	26.3 (23.8, 28.7)	
Prairie Provinces	3,064	25.2 (23.3, 27.0)	1,715	27.8 (24.8, 30.7)	3,413	32.0 (29.6, 34.3)*	4,669	36.6 (33.6, 39.6)*	
Males	1,341	30.4 (27.2, 33.5)	706	34.0 (29.0, 39.1)	1,428	37.0 (33.2, 40.8)*	2,040	42.6 (38.7, 46.5)*	
Females	1,723	20.6 (18.3, 22.8)	1,009	22.2 (18.9, 25.5)	1,985	27.5 (24.7, 30.3)*	2,629	31.3 (26.9, 35.7)*	
British Columbia	2,258	37.4 (34.5, 40.3)	627	33.5 (29.7, 37.3)	1,552	41.9 (38.2, 45.7)	2,092	41.9 (38.6, 45.3)*	
Males	1,013	40.6 (36.0, 45.1)	281	33.3 (27.6, 38.9)*	658	44.4 (38.8, 50.0)	923	46.7 (41.1, 52.2)	
Females	1,245	34.5 (31.0, 38.0)	346	33.7 (28.8, 38.7)	894	39.7 (34.4, 45.0)	1,169	37.7 (33.5, 41.9)	
Notes.

Data are presented as mean (95% CI).

* Significantly different from 2018, p <0.05.

Figure 1 Proportion of Canadians meeting physical activity guidelines across ages and regions.

Among adults 18–64 years at the national level, there was a significant reduction in the proportion of men (but not women) meeting the MVPA guidelines in both the fall of 2020 and 2021, when compared to 2018. In Québec, there was 7-percentage point reduction in the proportion of males meeting the recommendations in the fall of 2020, compared to 2018, while there was a 4-percentage point increase among females in 2021 (all p < 0.05). In Ontario and the Prairie provinces, males saw a 4-percentage point decrease in recommendation adherence in the fall of 2021, when compared to 2018 (p = 0.01). There were no other significant differences across regions when comparing the fall of 2020 or 2021 with 2018 (all p > 0.05).

Among adults aged 65+ years, there was a significant increase in the proportion of men meeting MVPA recommendations at the national level in the fall of 2020 (p < 0.05) and for both men and women in 2021 (p < 0.05). Similar trends were seen at the regional level. Compared to 2018, significant increases were observed in Atlantic Canada and the Prairies in the fall of 2020, and in Atlantic Canada, Québec and the Prairies in 2021 (all p < 0.05), although these changes were not always significant when examining men and women separately. Aside from an increase among Ontario men in the fall of 2020, there were no other changes observed in this age group for adults living in Ontario or BC (all p > 0.05).

Recreational MVPA

Daily recreational MVPA is reported in Table 4. At the national-level, youth reported an average decrease of 10 min per day of recreational MVPA in the fall of 2020, compared to 2018 (p < 0.001), while the difference between 2021 and 2018 was significant in girls, but not boys. Except for boys in Atlantic Canada and BC, all groups saw reductions in the fall of 2020 at the regional level (all p < 0.05). By the fall of 2021 most regions had returned to baseline levels, with the exception of Ontario, which remained lower when both genders were combined (but not in either gender individually).

Table 4 Minutes per day spent engaging in recreational physical activity before and during the COVID-19 pandemic.

	2018	Jan–Mar 2020	Sept–Dec 2020	2021	
	n	Estimate	n	Estimate	n	Estimate	n	Estimate	
12–17 years									
Canada-wide	3,952	29.8 (28.3, 31.4)	911	29.1 (26.3, 32.0)	1,573	20.3 (18.5, 22.1)*	3,501	27.6 (25.7, 29.6)	
Males	2,024	33.9 (31.6, 36.2)	465	31.5 (27.4, 35.5)	813	23.3 (20.5, 26.0)*	1,809	33.2 (30.3, 36.1)	
Females	1,928	25.4 (23.3, 27.5)	446	26.7 (22.6, 30.7)	760	17.2 (14.9, 19.5)*	1,692	21.8 (19.5, 24.2)*	
Atlantic Canada	496	31.0 (27.2, 34.7)	123	29.6 (22.6, 36.5)	170	24.1 (19.4, 28.8)*	422	33.2 (28.7, 37.7)	
Males	258	33.1 (27.3, 38.8)	62	33.3 (21.9, 44.6)	81	28.0 (20.0, 35.9)	217	35.0 (29.2, 40.8)	
Females	238	28.7 (24.3, 33.1)	61	25.4 (17.0, 33.9)	89	20.6 (15.3, 25.9)*	205	31.2 (24.3, 38.2)	
Québec	843	24.0 (21.0, 27.0)	206	28.1 (22.9, 33.3)	386	14.5 (11.8, 17.2)*	792	25.0 (20.8, 29.2)	
Males	417	29.3 (24.5, 34.0)	107	31.2 (23.4, 39.1)	190	16.6 (12.0, 21.3)*	393	32.0 (24.9, 39.0)	
Females	426	18.3 (14.9, 21.7)	99	24.9 (18.1, 31.7)	196	12.3 (9.1, 15.5)*	399	17.7 (13.7, 21.8)	
Ontario	1,185	31.3 (28.3, 34.3)	258	25.5 (20.1, 31.0)	413	21.0 (17.2, 24.7)*	1,003	26.4 (23.0, 29.9)*	
Males	618	35.7 (31.3, 40.0)	137	27.7 (20.4, 34.9)	212	22.8 (17.1, 28.6)*	526	31.2 (26.2, 36.2)	
Females	567	26.7 (22.7, 30.6)	121	23.2 (14.9, 31.6)	201	19.1 (14.2, 24.0*	477	21.5 (17.0, 25.9)	
Prairie Provinces	896	31.5 (28.4, 34.6)	196	29.9 (24.7, 35.2)	446	20.4 (17.4, 23.5)*	767	29.4 (25.8, 33.0)	
Males	458	36.4 (31.8, 40.9)	92	31.9 (23.9, 39.9)	245	24.2 (19.6, 28.9)*	391	34.8 (28.9, 40.7)	
Females	438	26.1 (21.8, 30.4)	104	28.0 (21.6, 34.5)	201	16.5 (12.6, 20.5)*	376	23.8 (19.9, 27.8)	
British Columbia	532	31.7 (27.4, 36.0)	128	41.1 (33.7, 48.5)*	158	25.8 (21.1, 30.5)	517	30.3 (26.3, 34.3)	
Males	273	32.7 (26.3, 39.1)	67	42.8 (33.0, 52.6)	85	32.4 (24.6, 40.1)	282	38.1 (32.3, 43.9)	
Females	259	30.6 (24.9, 36.3)	61	39.4 (28.5, 50.2)	73	18.9 (13.3, 24.5)*	235	22.2 (16.4, 27.9)	
18–64 years									
Canada-wide	34,040	15.0 (14.6, 15.4)	7,468	15.0 (14.1, 15.8)	11,459	15.4 (14.7, 16.2)	26,797	16.5 (16.0, 17.1)*	
Males	15,898	16.7 (16.1, 17.3)	3,485	16.4 (15.2, 17.7)	5,191	16.5 (15.3,17.6)	12,199	17.7 (16.8, 18.5)	
Females	18,142	13.3 (12.8, 13.8)	3,983	13.5 (12.4, 14.6)	6,268	14.4 (13.4, 15.4)*	14,598	15.4 (14.7,16.1)*	
Atlantic Canada	4,122	14.2 (13.3, 15.2)	792	15.8 (13.1, 18.6)	1,119	15.2 (13.1, 17.4)	5,346	17.2 (16.0, 18.3)*	
Males	1,800	14.3 (12.9, 15.8)	339	18.6 (13.8, 23.3)	510	16.4 (12.9, 19.9)	2,360	18.3 (16.4, 20.3)*	
Females	2,322	14.1 (12.9, 15.4)	453	13.2 (10.5, 15.9)	609	14.1 (11.7, 16.5)	2,986	16.0 (14.8, 17.2)*	
Québec	7,595	13.9 (13.1, 14.6)	1,609	14.8 (13.1, 16.6)	2,556	13.6 (12.1, 15.0)	3,914	15.5 (14.4, 16.7)*	
Males	3,642	15.5 (14.4, 16.5)	751	16.0 (13.5, 18.6)	1,192	14.7 (12.4, 17.1)	1,839	16.8 (15.0, 18.6)	
Females	3,953	12.2 (11.2, 13.2)	585	13.6 (11.6, 15.7)	1,364	12.3 (10.7, 14.0)	2,075	14.3 (12.9, 15.6)*	
Ontario	9,875	14.3 (13.6, 15.1)	2,514	13.9 (12.6, 15.3)	3,927	14.8 (13.5, 16.1)	7,182	15.3 (14.3, 16.3)	
Males	4,548	16.4 (15.2, 17.6)	1,186	16.1 (13.8, 18.3)	1,765	16.5 (14.5, 18.5)	3,216	16.4 (15.0, 17.8)	
Females	5,327	12.3 (11.5, 13.2)	1,328	11.8 (10.1, 13.5)	2,162	13.1 (11.5, 14.8)	3,966	14.2 (12.9, 15.6)*	
Prairie Provinces	7,742	15.8 (15.0, 16.6)	1,592	15.3 (13.4, 17.3)	2,474	15.9 (14.5, 17.4)	7,731	17.0 (16.0, 18.0)	
Males	3,705	17.4 (16.1, 18.7)	757	17.0 (14.3, 19.8)	1,134	16.6 (14.3, 18.8)	3,582	17.8 (16.3, 19.3)	
Females	4,037	14.2 (13.2, 15.1)	835	13.6 (11.3, 15.9)	1,340	15.3 (13.5, 17.1)	4,149	16.3 (15.1, 17.5)*	
British Columbia	4,706	18.2 (17.1, 19.3)	961	17.4 (14.9, 19.8)	1,383	19.9 (17.3, 22.4)	2,624	20.7 (18.9, 22.4)*	
Males	2,203	20.0 (18.2, 21.8)	452	16.2 (13.0, 19.5)*	590	19.1 (15.8, 22.5)	1,202	22.2 (19.7, 24.7)	
Females	2,503	16.4 (15.0, 17.8)	509	18.5 (14.6, 22.3)	792	20.6 (17.0, 24.2)*	1,422	19.2 (16.8, 21.6)*	
65+ years									
Canada-wide	16,053	9.0 (8.4, 9.6)	6,465	8.2 (7.4, 9.0)	14,202	9.2 (8.5, 9.8)	18,945	11.1 (10.5, 11.7)*	
Males	7,067	11.0 (9.9, 12.1)	2,821	10.0 (8.7, 11.3)	6,063	11.1 (10.0, 12.3)	8,254	13.0 (12.0, 14.0)*	
Females	8,986	7.3 (6.7, 7.8)	3,644	6.6 (5.5, 7.6)	8,139	7.4 (6.7, 8.2)	10,691	9.5 (8.7, 10.2)*	
Atlantic Canada	2,259	7.8 (6.8, 8.7)	1,859	7.4 (6.3, 8.5)	4,509	9.1 (8.1, 10.1)	4,113	8.7 (7.8, 9.7)	
Males	1,002	9.7 (8.0, 11.4)	824	8.1 (6.1, 10.1)	1,933	10.3 (8.7, 11.9)	1,788	11.5 (9.7, 13.3)	
Females	1,257	6.1 (5.0, 7.1)	1,035	6.7 (5.3, 8.2)	2,576	8.0 (6.7, 9.4)*	2,325	6.3 (5.5, 7.1)	
Québec	3,582	8.1 (7.1, 9.1)	907	7.9 (6.0, 9.8)	1,762	6.9 (5.7, 8.0)	2,641	9.9 (8.8, 11.1)*	
Males	1,613	10.1 (8.4, 11.8)	419	7.6 (5.6, 9.5)	779	7.7 (5.9, 9.5)	1,156	12.6 (10.5, 14.6)	
Females	1,969	6.5 (5.5, 7.4)	488	8.2 (5.2, 11.2)	983	6.1 (4.7, 7.5)	1,485	7.6 (6.4, 8.8)	
Ontario	4,890	9.2 (8.0, 10.4)	1,357	7.2 (5.8, 8.5)*	2,966	9.2 (7.8, 10.7)	5,430	10.6 (9.5, 11.7)	
Males	2,098	11.1 (8.8, 13.5)	591	10.4 (7.7, 13.0)	1,265	11.7 (9.4, 14.1)	2,347	11.9 (10.2, 13.6)	
Females	2,792	7.5 (6.4, 8.6)	766	4.4 (3.3, 5.6)*	1,701	7.1 (5.5, 8.7)	3,083	9.5 (8.1, 11.0)*	
Prairie Provinces	3,064	8.4 (7.4, 9.4)	1,715	10.2 (8.5, 11.9)	3,413	10.8 (9.4, 12.3)*	4,669	11.9 (10.5, 13.4)*	
Males	1,341	10.2 (8.5, 11.9)	706	13.1 (10.3, 15.9)	1,428	13.0 (10.4, 15.6)	2,040	12.5 (10.5, 14.6)	
Females	1,723	6.9 (5.8, 8.0)	1,009	7.7 (5.8, 9.6)	1,985	8.9 (7.5, 10.4)*	2,629	11.4 (9.4, 13.4)*	
British Columbia	2,258	11.4 (10.1, 12.7)	627	9.7 (7.6, 11.8)	1,552	11.2 (9.5, 12.8)	2,092	14.8 (13.2, 16.5)*	
Males	1,013	13.9 (11.8, 16.0)	281	11.2 (7.8, 14.6)	658	13.8 (11.0, 16.5)	923	17.8 (15.0, 20.6)*	
Females	1,245	9.1 (7.6, 10.6)	346	8.3 (5.5, 11.0)	894	8.8 (7.1, 10.5)	1,169	12.2 (10.3, 14.0)*	
Notes.

Data are presented as mean (95% CI).

* Significantly different from 2018, p <0.05.

Among adults aged 18–64 years, recreational MVPA levels generally remained stable between 2018 and the fall of 2020, with the exception of increased MVPA among women in BC (p < 0.05). In the fall of 2021, significant increases in recreational MVPA were seen nationally, as well as in Québec (overall and in women) and in, Ontario, and the Prairie Provinces (women only) (all p < 0.05). Among males, an increase in recreational MVPA was seen only in Atlantic Canada. Among adults 65+ years, recreational MVPA levels were significantly increased in the fall of 2020 in females in Atlantic Canada and the Prairie Provinces (all p <  0.05). In the fall of 2021, increases were observed for females in Ontario and the Prairies, in both genders in BC, and overall in Québec (but not in either gender individually).

Active transportation

Active transportation decreased significantly among boys and girls at the national level in the fall of 2020 (all p < 0.001) and remained at this level in the fall of 2021 for girls, but not boys (Table 5). Both genders in Ontario and the Prairies, and boys in Québec saw reduced time spent in active transportation in the fall of 2020, which remained reduced in 2021 in Ontario only (all p < 0.05). There was no change in active transportation in either Atlantic Canada or BC at any time point (all p > 0.05). Among adults 18–64 years nation-wide, there was a significant reduction in active transportation levels in the fall of 2020 for females, and for both genders in 2021 (all p < 0.05). At the regional level in the fall of 2020, a significant increase was seen for females in Atlantic Canada, along with a significant decrease for females in Ontario, with no changes seen among males in any region. In the fall of 2021, decrease in active transportation was observed in BC (among women), Ontario saw a decrease in both genders, and Atlantic Canada saw an increase among males only. Among adults 65+ years there were small but significant increases nation-wide in the fall of 2020 but not 2021 (all p < 0.001) for both males and females. At the regional level, overall increases were seen in the fall of 2020 in Ontario, and for both males and females in Atlantic Canada (all p < 0.05). In the fall of 2021, Atlantic Canada, Quebec and the Prairies saw an increase either overall or in at least one gender, while BC saw an overall decrease (all p < 0.05). There were no changes seen in Ontario when comparing 2021 and 2018 (p > 0.05).

Table 5 Minutes per day spent engaging in active transportation before and during the COVID-19 pandemic.

	2018	Jan–Mar 2020	Sept–Dec 2020	2021	
	n	Estimate	n	Estimate	n	Estimate	n	Estimate	
12–17 years									
Canada-wide	3,952	24.8 (23.3, 26.3)	911	23.1 (20.4, 25.7)	1,573	19.6 (17.8, 21.4)*	3,501	20.4 (18.8, 22.0)*	
Males	2,024	25.9 (23.6, 28.1)	465	24.7 (20.6, 28.9)	813	19.4 (16.9, 21.9)*	1,809	22.7 (20.1, 25.4)	
Females	1,928	23.6 (21.5, 25.7)	446	21.3 (18.2, 24.4)	760	19.8 (17.2, 22.3*	1,692	17.9(16.3, 19.6)*	
Atlantic Canada	496	17.3 (14.2, 20.4)	123	21.8 (15.0, 28.6)	170	19.7 (14.6, 24.9)	422	18.8 (15.4, 22.3)	
Males	258	18.6 (14.3, 22.9)	62	25.6E (15.9, 35.4)	81	19.6E (11.7, 27.5)	217	21.0 (15.5, 26.6)	
Females	238	15.9 (11.4, 20.3)	61	17.9E (9.0, 26.7)	89	19.8E (12.9, 26.8)	205	16.5 (12.2, 20.8)	
Québec	843	21.5 (18.8, 24.2)	206	18.7 (14.7, 22.6)	386	15.8 (13.2, 18.4)*	792	21.9 (18.0, 25.8)	
Males	417	22.1 (18.2, 25.9)	107	17.9 (12.5, 23.2)	190	13.3 (10.5, 16.1)*	393	23.8 (17.1, 30.6)	
Females	426	20.8 (17.0, 24.7)	99	19.4 (13.3, 25.6)	196	18.4 (13.8, 22.9)	399	19.9 (16.2, 23.5)	
Ontario	1,185	27.3 (24.3, 30.4)	258	23.9 (18.3, 29.5)	413	19.7 (16.3, 23.1)*	1,003	17.1 (14.7, 19.5)*	
Males	618	28.1 (23.6, 32.5)	137	29.1 (19.6, 38.5)	212	19.3 (14.3, 24.2)*	526	20.1 (16.1, 24.1)*	
Females	567	26.6 (22.5, 30.7)	121	18.2 (12.8, 23.7)*	201	20.1 (15.4, 24.8*	477	14.0 (11.3, 16.8)*	
Prairie Provinces	896	24.1 (21.2, 27.1)	196	21.1 (16.9, 25.3)	446	17.3 (14.5, 20.0)*	767	23.8 (19.8, 27.7)	
Males	458	25.7 (21.1, 30.3)	92	20.5 (14.7, 26.4)	245	17.9 (14.2, 21.6)*	391	24.9 (18.5, 31.2)	
Females	438	22.4 (18.8, 26.0)	104	21.6 (15.9, 27.4)	201	16.6 (12.4, 20.8*	376	22.6 (18.3, 26.8)	
British Columbia	532	26.6 (23.0, 30.2)	128	32.2 (25.8, 38.6)	158	29.2 (22.8, 35.7)	517	23.8 (20.6, 26.9)	
Males	273	29.0 (23.6, 34.3)	67	29.1 (21.7, 36.4)	85	32.0 (22.8, 41.1)	282	26.7 (21.5, 31.8)	
Females	259	24.0 (19.0, 29.0)	61	35.4 (25.2, 45.7)	73	26.3 (17.6, 34.9)	235	20.7 (16.8, 24.7)	
18–64 years									
Canada-wide	34,040	13.8 (13.4, 14.3)	7,468	13.3 (12.5, 14.2)	11,459	12.9 (12.2, 13.7)*	26,797	12.3 (11.8, 12.7)*	
Males	15,898	14.7 (14.0, 15.4)	3,485	14.9 (13.6, 16.2)	5,191	14.1 (13.0, 15.2)	12,199	12.8 (12.1, 13.5)*	
Females	18,142	13.0 (12.4, 13.5)	3,983	11.7 (10.7, 12.8)*	6,268	11.8 (10.8, 12.7)*	14,598	11.7 (11.0, 12.4)*	
Atlantic Canada	4,122	8.8 (7.9, 9.6)	792	14.7 (10.9, 18.5)*	1,119	11.0 (8.9, 13.1)	5,346	10.0 (9.1, 10.9)*	
Males	1,800	9.3 (8.0, 10.6)	339	18.3E (11.9, 24.7)*	510	9.7 (7.2, 12.3)	2,360	11.7 (10.2, 13.2)*	
Females	2,322	8.2 (7.3, 9.2)	453	11.3E (7.4, 15.2)	609	12.2 (9.1, 15.3)*	2,986	8.4 (7.4, 9.3)	
Québec	7,595	11.8 (11.1, 12.6)	1,609	12.3 (10.7, 14.0)	2,556	11.1 (9.5, 12.7)	3,914	12.2 (11.2, 13.2)	
Males	3,642	12.9 (11.6, 14.2)	751	14.5 (11.5, 17.5)	1,192	12.1 (9.8, 14.3)	1,839	12.3 (10.8, 13.9)	
Females	3,953	10.8 (9.9, 11.6)	585	10.0 (8.5, 11.6)	1,364	10.1 (8.0, 12.2)	2,075	12.1 (10.8, 13.5)	
Ontario	9,875	15.2 (14.3, 16.2)	2,514	12.0 (10.8, 13.2)*	3,927	13.0 (11.8, 14.3)*	7,182	11.9 (11.0, 12.8)*	
Males	4,548	16.3 (14.9, 17.7)	1,186	12.6 (10.8, 14.5)*	1,765	14.1 (12.2, 16.0)	3,216	12.1 (10.8, 13.4)*	
Females	5,327	14.2 (13.1, 15.4)	1,328	11.4 (10.1, 12.8)*	2,162	12.0 (10.3, 13.7)*	3,966	11.8 (10.5, 13.1)*	
Prairie Provinces	7,742	11.9 (11.1, 12.6)	1,592	13.4 (11.6, 15.3)	2,474	11.6 (10.4, 12.9)	7,731	11.4 (10.6, 12.3)	
Males	3,705	12.9 (11.7, 14.1)	757	16.5 (13.7, 19.3)*	1,134	13.4 (11.3, 15.5)	3,582	12.2 (11.0, 13.5)	
Females	4,037	10.9 (10.0, 11.7)	835	10.4 (7.9, 12.8)	1,340	9.8 (8.5, 11.1)	4,149	10.6 (9.5, 11.7)	
British Columbia	4,706	18.4 (17.1, 19.7)	961	18.2 (15.8, 20.6)	1,383	18.5 (16.0, 21.1)	2,624	15.2 (13.8, 16.7)*	
Males	2,203	18.5 (16.5, 20.5)	452	18.9 (15.3, 22.5)	590	20.8 (16.5, 25.0)	1,202	16.9 (14.7, 19.1)	
Females	2,503	18.2 (16.6, 19.8)	509	17.5 (14.3, 20.7)	792	16.4 (13.9, 18.8)	1,422	13.6 (11.8, 15.4)*	
65+ years									
Canada-wide	16,053	9.3 (8.7, 9.9)	6,465	11.7 (10.7, 12.6)*	14,202	11.1 (10.3, 11.9)*	18,945	10.0 (9.4, 10.5)	
Males	7,067	10.0 (9.1, 10.9)	2,821	13.0 (11.5, 14.5)*	6,063	12.1 (10.8, 13.4)*	8,254	10.7 (9.8, 11.5)	
Females	8,986	8.7 (8.0, 9.4)	3,644	10.5 (9.3, 11.7)*	8,139	10.3 (9.3, 11.3)*	10,691	9.4 (8.6, 10.1)	
Atlantic Canada	2,259	5.1 (4.3, 5.9)	1,859	7.5 (6.2, 8.7)*	4,509	7.9 (7.0, 8.8)*	4,113	7.1 (6.2, 7.9)*	
Males	1,002	5.4 (4.2, 6.6)	824	8.8 (6.9, 10.6)*	1,933	9.7 (8.1, 11.2)*	1,788	8.5 (7.0, 10.0)*	
Females	1,257	4.8 (3.7, 5.9)	1,035	6.3 (4.8, 7.8)	2,576	6.4 (5.4, 7.4)*	2,325	5.8 (4.9, 6.6)	
Québec	3,582	7.4 (6.6, 8.2)	907	10.2 (8.6, 11.8)*	1,762	7.7 (6.5, 8.9)	2,641	9.4 (8.4, 10.4)*	
Males	1,613	8.0 (6.7, 9.3)	419	10.9 (7.9, 13.8)	779	8.5 (6.4, 10.7)	1,156	10.2 (8.6, 11.7)*	
Females	1,969	6.9 (5.9, 7.8)	488	9.6 (7.5, 11.7)*	983	7.0 (5.6, 8.5)	1,485	8.6 (7.4, 9.9)*	
Ontario	4,890	9.4 (8.2, 10.5)	1,357	12.1 (10.3, 13.9)*	2,966	11.8 (10.1, 13.5)*	5,430	9.2 (8.3, 10.2)	
Males	2,098	10.4 (8.6, 12.2)	591	14.7 (11.7, 17.8)*	1,265	12.7 (10.1, 15.4)	2,347	9.3 (7.9, 10.7)	
Females	2,792	8.5 (7.1, 9.8)	766	9.9 (7.8, 12.0)	1,701	11.0 (8.9, 13.1)	3,083	9.1 (7.8, 10.5)	
Prairie Provinces	3,064	8.3 (7.1, 9.4)	1,715	10.4 (8.5, 12.4)	3,413	9.7 (8.5, 10.8)	4,669	11.0 (9.5, 12.6)*	
Males	1,341	9.1 (7.4, 10.8)	706	13.3 (9.6, 16.9)*	1,428	10.9 (9.0, 12.7)	2,040	13.0 (11.1, 14.8)*	
Females	1,723	7.5 (6.0, 9.0)	1,009	8.0 (6.3, 9.7)	1,985	8.6 (7.1, 10.1)	2,629	9.3 (7.0, 11.6)	
British Columbia	2,258	16.1 (14.3, 17.8)	627	16.6 (13.8, 19.4)	1,552	18.7 (16.4, 21.1)	2,092	13.5 (11.8, 15.3)*	
Males	1,013	15.8 (13.3, 18.4)	281	14.0 (10.2, 17.9)	658	19.1 (15.6, 22.5)	923	13.8 (11.0, 16.5)	
Females	1,245	16.3 (14.1, 18.5)	346	18.9 (14.9, 22.9)	894	18.4 (15.3, 21.5)	1,169	13.3 (11.2, 15.4)	
Notes.

Data are presented as mean (95% CI).

* Significantly different from 2018, p <0.05.

E interpret with caution, co-efficient of variation >16.6.

School-related MVPA

Nationally, school-related MVPA was significantly lower in the fall of 2020 and 2021 when compared to 2018 for both genders (all p < 0.05) (Table 6). School-related MVPA was significantly reduced in the fall of 2020 for boys in the Prairie Provinces, girls in BC, and both genders in Ontario (all p < 0.05). In the fall of 2021, school-related MVPA was significantly reduced from 2018 levels in Québec females and both genders in Ontario, with no significant changes seen in any other groups.

Table 6 Minutes per day spent engaging in school-related physical activity before and during the COVID-19 pandemic.

	2018	Jan–Mar 2020	Sept–Dec 2020	2021	
	n	Estimate	n	Estimate	n	Estimate	n	Estimate	
12–17 years									
Canada-wide	3,952	19.1 (18.0, 20.3)	911	22.0 (19.9, 24.0)*	1,573	13.0 (11.6, 14.5)*	3,501	13.8 (12.8, 14.9)*	
Males	2,024	21.1 (19.3, 22.8)	465	23.6 (20.7, 26.5)	813	14.6 (12.4, 16.8)*	1,809	16.4 (14.7, 18.2)*	
Females	1,928	17.0 (15.4, 18.6)	446	20.2 (17.4, 22.9)	760	11.4 (9.5, 13.3)*	1,692	11.2 (10.0, 12.4)*	
Atlantic Canada	496	16.2 (13.5, 18.8)	123	16.0 (12.2, 19.8)	170	12.1 (8.8, 15.4)	422	13.0 (10.6, 15.4)	
Males	258	18.0 (14.1, 21.9)	62	15.9E (10.7, 21.1)	81	12.0E (6.8, 17.1)	217	14.8 (11.2, 18.3)	
Females	238	14.1 (10.7, 17.5)	61	16.2E (10.6, 21.8)	89	12.2E (7.9, 16.5)	205	11.1 (7.8, 14.4)	
Québec	843	19.4 (17.1, 21.8)	206	23.3 (19.4, 27.2)	386	17.6 (14.6, 20.6)	792	13.7 (11.5, 16.0)*	
Males	417	19.5 (16.7, 22.4)	107	21.5 (15.7, 27.2)	190	19.8 (15.5, 24.2)	393	14.9 (11.0, 18.7)	
Females	426	19.4 (15.6, 23.2)	99	25.2 (20.1, 30.2)	196	15.4 (11.3, 19.5)	399	12.6 (10.1, 15.1)*	
Ontario	1,185	18.5 (16.3, 20.6)	258	20.8 (17.0, 24.6)	413	10.3 (7.6, 12.9)*	1,003	9.8 (8.1, 11.5)*	
Males	618	20.6 (17.4, 23.8)	137	24.5 (18.9, 30.0)	212	10.2E (6.4, 13.9)*	526	13.1 (10.3, 16.0)*	
Females	567	16.2 (13.5, 19.0)	121	16.8 (11.8, 21.8)	201	10.4E (6.7, 14.0)*	477	6.4 (4.7, 8.0)*	
Prairie Provinces	896	20.5 (18.0, 23.0)	196	23.7 (19.6, 27.9)	446	14.3 (11.7, 17.0)*	767	18.6 (16.1, 21.2)	
Males	458	23.8 (19.9, 27.6)	92	23.1 (18.1, 28.2)	245	16.0 (12.0, 20.0)*	391	20.6 (16.6, 24.6)	
Females	438	16.9 (14.0, 19.9)	104	24.4 (17.7, 31.1)*	201	12.6 (8.9, 16.3)	376	16.6 (13.4, 19.8)	
British Columbia	532	20.1 (17.0, 23.1)	128	23.5 (18.6, 28.5)	158	12.5 (8.8, 16.1)*	517	19.6 (16.2, 23.0)	
Males	273	22.6 (17.8, 27.5)	67	29.3 (22.6, 36.0)	85	18.8E (12.3, 25.3)	282	23.5 (18.1, 28.9)	
Females	259	17.4 (13.9, 20.9)	61	17.5 (10.1, 24.9)	73	5.7E (2.6, 8.8)*	235	15.7 (11.8, 19.6)	
Notes.

Data are presented as mean (95% CI).

* Significantly different from 2018, p <0.05.

E interpret with caution, co-efficient of variation >16.6.

Occupational/household MVPA

Although there was excessive variability to perform statistical testing for examine occupational/household MVPA at the regional level among youth, nationally this form of activity increased in 2021 (p < 0.001), but not the fall of 2020 (p =0.71), when compared to 2018 in this age group (Table 7). Among adults 18–64 years national trends saw no change in the fall of 2020, with a significant decrease seen in males only (p <  0.05). Atlantic Canada observed a significant decrease among women in the fall of 2020, and among both genders in the fall of 2021 (all p < 0.05). Ontario also saw a significant reduction among males in the fall of 2021 (all p <  0.05). There were no other significant changes observed in any region for this age group (all p > 0.05). Among adults 65+ years, the national trend was for a significant increase in household and occupational MVPA for both genders in both the fall of 2020 and in 2021 (all p < 0.05). At the regional level, significant increases in the fall of 2020 were seen among females in Québec and the Prairie Provinces, and among males in Ontario (all p < 0.05). In the fall of 2021, we observed significant increases in occupational and household MVPA for both males and females in the Prairie Provinces, females in Québec, and for BC overall (but not in either gender individually) (all p < 0.05).

Table 7 Minutes per day spent engaging in household and occupational physical activity before and during the COVID-19 pandemic.

	2018	Jan–Mar 2020	Sept–Dec 2020	2021	
	n	Estimate	n	Estimate	n	Estimate	n	Estimate	
12–17 years									
Canada-wide	3,952	4.9 (4.3, 5.4)	911	4.6E (3.0, 6.2)	1,573	5.1 (3.9, 6.4)	3,501	6.9 (5.9, 7.9)*	
Males	2,024	5.4 (4.6, 6.2)	465	6.0E (3.2, 8.7)	813	5.7E (3.7, 7.7)	1,809	7.7 (6.1, 9.4)*	
Females	1,928	4.3 (3.5, 5.1)	446	3.2E (1.5, 4.8)	760	4.5 (3.0, 5.9)	1,692	6.0 (4.8, 7.2)*	
Atlantic Canada	496	4.7E (2.8, 6.6)	123	4.3E (1.8, 6.7)	170	–	422	6.4E (4.0, 8.8)	
Males	258	5.2E (2.6, 7.8)	62	–	81	–	217	7.8E (4.1, 11.5)	
Females	238	4.2E (1.7, 6.6)	61	–	89	–	205	5.0E (2.0, 8.0)	
Québec	843	4.5 (3.3, 5.7)	206	–	386	3.2E (1.4, 4.9)	792	7.1E (4.4, 9.8)	
Males	417	6.6 (4.5, 8.7)	107	1.8 (0.6, 2.9)	190	–	393	9.1E (4.3, 14.0)	
Females	426	2.3E (1.3, 3.4)	99	–	196	3.6E (1.5, 5.6)	399	5.1E (2.8, 7.3)	
Ontario	1,185	3.8 (3.0, 4.7)	258	6.2E (2.5, 9.9)	413	5.7E (3.1, 8.3)	1,003	5.5 (4.2, 6.9)	
Males	618	4.1 (2.9, 5.4)	137	–	212	6.9E (2.7, 11.2)	526	5.2 (3.8, 6.7)	
Females	567	3.5E (2.4, 4.7)	121	–	201	–	477	5.9E (3.7, 8.0)	
Prairie Provinces	896	6.9 (5.4, 8.4)	196	4.6E (2.2, 7.1)	446	4.3 (3.0, 5.7)	767	10.3 (7.4, 13.2)	
Males	458	5.5 (3.8, 7.1)	92	7.3E (2.7, 12.0)	245	4.2E (2.3, 6.1)	391	11.9E (7.4, 16.4)	
Females	438	8.5 (5.8, 11.2)	104	–	201	4.4E (2.6, 6.3)	376	8.6 (5.1, 12.1)	
British Columbia	532	5.6 (4.0, 7.1)	128	3.0E (1.2, 4.9)	158	8.2E (4.4, 11.9)	517	5.5 (3.5, 7.4)	
Males	273	7.6E (5.1, 10.1)	67	–	85	8.9E (3.1, 14.7)	282	6.4E (3.8, 9.0)	
Females	259	3.5 (1.6, 5.4)	61	–	73	7.3E (2.8, 11.8)	235	4.5E (1.7, 7.4)	
18–64 years									
Canada-wide	34,040	16.6 (16.0, 17.2)	7,468	13.5 (12.6, 14.5)*	11,459	15.9 (14.9, 16.8)	26,797	16.0 (15.4, 16.6)	
Males	15,898	20.8 (19.8, 21.7)	3,485	16.5 (14.9, 18.1)*	5,191	19.4 (17.8, 21.1)	12,199	19.2 (18.3, 20.2)*	
Females	18,142	12.4 (11.8, 13.0)	3,983	10.6 (9.4, 11.7)*	6,268	12.3 (11.3, 13.2)	14,598	12.7 (11.9, 13.5)	
Atlantic Canada	4,122	23.2 (21.5, 24.8)	792	19.3 (15.8, 22.8)	1,119	18.7 (16.0, 21.5)*	5,346	18.8 (17.5, 20.2)*	
Males	1,800	27.7 (25.1, 30.3)	339	24.9 (19.4, 30.4)	510	22.6 (17.6, 27.5)	2,360	22.5 (20.4, 24.7)*	
Females	2,322	18.8 (16.8, 20.7)	453	13.9 (10.0, 17.8)*	609	15.1 (12.5, 17.7)*	2,986	15.3 (13.7, 16.8)*	
Québec	7,595	13.7 (12.7, 14.8)	1,609	12.9 (11.0, 14.8)	2,556	13.2 (11.4, 15.0)	3,914	13.4 (12.1, 14.7)	
Males	3,642	18.0 (16.4, 19.7)	751	18.5 (15.0, 22.0)	1,192	16.9 (13.9, 19.9)	1,839	17.3 (15.1, 19.6)	
Females	3,953	9.3 (8.1, 10.5)	585	7.2 (5.7, 8.6)*	1,364	9.3 (7.4, 11.3)	2,075	9.4 (8.1, 10.6)	
Ontario	9,875	15.4 (14.4, 16.4)	2,514	11.8 (10.3, 13.4)*	3,927	14.7 (13.1, 16.2)	7,182	14.7 (13.7, 15.8)	
Males	4,548	19.6 (17.9, 21.3)	1,186	13.4 (10.8, 16.1)*	1,765	18.3 (15.6, 21.0)	3,216	16.5 (15.1, 18.0)*	
Females	5,327	11.3 (10.3, 12.3)	1,328	10.3 (8.4, 12.2)	2,162	11.1 (9.4, 12.7)	3,966	13.0 (11.4, 14.6)	
Prairie Provinces	7,742	19.7 (18.5, 21.0)	1,592	14.8 (12.6, 17.0)*	2,474	17.9 (16.1, 19.6)	7,731	18.9 (17.7, 20.1)	
Males	3,705	23.9 (21.9, 25.9)	757	16.5 (13.3, 19.7)*	1,134	20.9 (17.9, 23.8)	3,582	23.0 (21.1, 24.9)	
Females	4,037	15.4 (3.8, 16.9)	835	13.0 (10.2, 15.9)	1,340	14.8 (12.7, 17.0)	4,149	14.7 (13.3, 16.0)	
British Columbia	4,706	17.5 (16.1, 18.9)	961	15.3 (12.6, 17.9)	1,383	19.8 (16.9, 22.6)	2,624	18.4 (16.5, 20.4)	
Males	2,203	21.2 (19.0, 23.4)	452	18.6 (14.0, 23.1)	590	23.8 (18.6, 29.0)	1,202	23.4 (20.1, 26.6)	
Females	2,503	13.8 (12.1, 15.5)	509	12.0 (8.6, 15.4)	792	15.8 (12.8, 18.8)	1,422	13.6 (11.6, 15.7)	
65+ years									
Canada-wide	16,053	8.4 (7.8, 8.9)	6,465	6.5 (5.7, 7.3)*	14,202	10.1 (9.4, 10.8)*	18,945	10.2 (9.6, 10.8)*	
Males	7,067	9.9 (9.1, 10.8)	2,821	7.4 (6.3, 8.6)*	6,063	12.2 (11.0, 13.4)*	8,254	11.8 (10.9, 12.7)*	
Females	8,986	7.0 (6.2, 7.8)	3,644	5.7 (4.7, 6.7)	8,139	8.2 (7.4, 9.1)*	10,691	8.8 (8.0, 9.6)*	
Atlantic Canada	2,259	11.5 (9.9, 13.1)	1,859	9.8 (8.4, 11.1)	4,509	11.5 (10.3, 12.8)	4,113	12.2 (10.9, 13.5)	
Males	1,002	15.0 (12.4, 17.5)	824	11.5 (9.4, 13.6)*	1,933	14.7 (12.6, 16.8)	1,788	15.6 (13.3, 17.9)	
Females	1,257	8.4 (6.6, 10.3)	1,035	8.3 (6.4, 10.2)	2,576	8.8 (7.4, 10.3)	2,325	9.2 (7.7, 10.7)	
Québec	3,582	6.3 (5.5, 7.0)	907	5.7 (4.2, 7.3)	1,762	7.6 (6.3, 8.8)	2,641	7.7 (6.6, 8.8)*	
Males	1,613	8.5 (7.2, 9.9)	419	6.6E (4.1, 9.0)	779	8.9 (6.6, 11.1)	1,156	8.5 (6.8, 10.2)	
Females	1,969	4.3 (3.5, 5.1)	488	5.0E (3.2, 6.7)	983	6.4 (4.9, 8.0)*	1,485	6.9 (5.5, 8.3)*	
Ontario	4,890	8.2 (7.0, 9.4)	1,357	5.8 (4.3, 7.3)*	2,966	10.0 (8.6, 11.4)	5,430	8.6 (7.7, 9.5)	
Males	2,098	8.7 (7.1, 10.3)	591	6.8 (4.8, 8.8)	1,265	13.7 (11.2, 16.2)*	2,347	9.6 (8.2, 11.0)	
Females	2,792	7.7 (6.0, 9.4)	766	4.9E (2.9, 7.0)*	1,701	6.9 (5.4, 8.4)	3,083	7.8 (6.5, 9.0)	
Prairie Provinces	3,064	8.8 (7.6, 10.0)	1,715	7.2 (5.5, 8.9)	3,413	11.7 (10.3, 13.1)*	4,669	13.9 (12.3, 15.6)*	
Males	1,341	11.4 (9.2, 13.5)	706	7.9 (5.6, 10.1)*	1,428	13.4 (11.2, 15.5)	2,040	17.5 (14.8, 20.3)*	
Females	1,723	6.5 (5.4, 7.6)	1,009	6.6 (4.1, 9.1)	1,985	10.2 (8.4, 12.0)*	2,629	10.8 (8.9, 12.7)*	
British Columbia	2,258	10.4 (8.8, 11.9)	627	7.4 (5.7, 9.1)*	1,552	12.3 (10.2, 14.4)	2,092	13.7 (11.7, 15.8)*	
Males	1,013	11.3 (9.1, 13.5)	281	8.2 (5.4, 10.9)	658	11.8 (9.1, 14.6)	923	15.3 (11.8, 18.7)	
Females	1,245	9.5 (7.4, 11.7)	346	6.7 (4.8, 8.6)	894	12.7 (9.4, 16.0)	1,169	12.4 (9.9, 14.8)	
Notes.

Data are presented as mean (95% CI).

E interpret with caution, co-efficient of variation >16.6. Empty cells not reported due to co-efficient of variation >33.3.

* Significantly different from 2018, p <0.05.

Recreational screen time

Compared to 2018, the proportion of youth accumulating ≤2 h per day of recreational screen time per day in 2021 decreased significantly at the national level for both genders on both school days and non-school days (all p < 0.05) (Table 8, Fig. 2). Females in BC, and both genders in Ontario and Atlantic Canada were significantly less likely to meet recommendations on week- or weekend-days, while females in the Prairie Provinces were less likely to meet recommendations on weekends only.

Table 8 Proportion of participants reporting ≤ 2 hours/day of recreational screen use before and during the COVID-19 pandemic.

	Work/school days	Non-work/non-school days	
	2018	2021	2018	2021	
	n	Estimate	n	Estimate	n	Estimate	n	Estimate	
12–17 years									
Canada-wide	3,952	40.7 (38.4, 43.0)	3,501	29.1 (26.9, 31.5)*	3,952	21.4 (19.7, 23.2)	3,501	13.2 (11.6, 15.0)*	
Males	2,024	37.8 (34.7, 40.9)	1,809	28.6 (25.4, 31.9)*	2,024	20.1 (17.8, 22.6)	1,809	14.0 (11.7, 16.8)*	
Females	1,928	43.8 (40.4, 47.2)	1,692	29.8 (26.7, 33.1)*	1,928	22.8 (20.3, 25.5)	1,692	12.3 (10.2, 14.8)*	
Atlantic Canada	496	46.8 (41.3, 52.4)	422	32.8 (26.9, 39.2)*	496	22.6 (18.5, 27.2)	422	12.3 (9.0, 16.4)*	
Males	258	48.5 (40.8, 56.2)	217	34.9 (26.6, 44.2)*	258	22.9 (17.2, 29.8)	217	14.4 (9.7, 20.9)*	
Females	238	45.1 (37.4, 53.1)	205	30.7 (22.9, 39.8)*	238	22.2 (16.8, 28.8)	205	9.9 (6.0, 15.8)*	
Québec	843	37.2 (32.7, 41.9)	792	34.8 (30.2, 39.7)	843	17.3 (14.6, 20.5)	792	14.3 (11.3, 18.0)	
Males	417	36.0 (30.0, 42.4)	393	30.8 (24.6, 37.7)	417	15.6 (11.9, 20.2)	393	13.7 (9.7, 19.0)	
Females	426	38.5 (32.2, 45.2)	399	39.0 (32.1, 46.4)	426	19.1 (15.1, 23.8)	399	14.9 (10.7, 20.5)	
Ontario	1,185	40.2 (36.0, 44.6)	1,003	22.6 (18.7, 27.1)*	1,185	22.1 (18.9, 25.6)	1,003	11.1 (8.3, 14.6)*	
Males	618	36.3 (30.5, 42.4)	526	23.4 (17.8, 30.2)*	618	21.5 (17.0, 26.7)	526	13.1 (9.0, 18.6)*	
Females	567	44.2 (38.0, 50.6)	477	21.8 (16.7, 27.9)*	567	22.7 (18.3, 27.9)	477	9.0 (5.7, 13.9)*	
Prairie Provinces	896	42.6 (38.4, 47.0)	767	34.7 (29.9, 39.9)*	896	24.3 (21.0, 28.0)	767	17.1 (13.6, 21.2)*	
Males	458	41.0 (35.3, 46.9)	391	32.6 (26.2, 39.7)	458	22.9 (18.6, 28.0)	391	16.5 (12.0, 22.3)	
Females	438	44.4 (38.0, 51.0)	376	37.0 (30.0, 44.6)	438	25.8 (20.9, 31.3)	376	17.7 (13.1, 23.7)*	
British Columbia	532	42.5 (36.6, 48.6)	517	29.8 (24.8, 35.3)*	532	21.0 (16.8, 26.0)	517	12.2 (9.4, 15.7)*	
Males	273	35.7 (28.2, 44.1)	282	32.0 (24.9, 40.1)	273	17.6 (12.4, 24.4)	282	13.7 (9.6, 19.0)	
Females	259	49.6 (40.8, 58.3)	235	27.7 (21.3, 35.1)*	259	24.7 (18.4, 32.3)	235	10.8 (7.4, 15.5)*	
18–64 years									
Canada-wide	34,040	53.9 (52.9, 55.0)	26,797	45.0 (43.9, 46.2)*	34,040	37.8 (36.9, 38.7)	26,797	28.0 (27.1, 29.0)*	
Males	15,898	50.7 (49.3, 52.1)	12,199	42.9 (41.3, 44.6)*	15,898	35.5 (34.2, 36.7)	12,199	26.2 (24.9, 27.6)*	
Females	18,142	57.6 (56.2, 59.0)	14,598	47.3 (45.7, 49.0)*	18,142	40.1 (38.9, 41.4)	14,598	29.8 (28.5, 31.2)*	
Atlantic Canada	4,122	58.8 (56.3, 61.3)	5,346	49.1 (46.8, 51.4)*	4,122	37.6 (35.7, 39.6)	5,346	30.2 (28.4, 32.1)*	
Males	1,800	55.9 (52.3, 59.4)	2,360	44.7 (41.3, 48.1)*	1,800	36.2 (33.4, 39.0)	2,360	28.4 (25.9, 31.1)*	
Females	2,322	62.0 (58.7, 65.1)	2,986	53.8 (50.8, 56.9)*	2,322	39.0 (36.2, 41.8)	2,986	32.0 (29.6, 34.4)*	
Québec	7,595	56.3 (54.4, 58.2)	3,914	46.6 (44.0, 49.1)*	7,595	36.6 (34.9, 38.3)	3,914	30.6 (28.5, 32.7)*	
Males	3,642	54.0 (51.2, 56.7)	1,839	43.6 (39.8, 47.4)*	3,642	35.8 (33.4, 38.3)	1,839	29.2 (26.2, 32.4)*	
Females	3,953	58.8 (56.2, 61.4)	2,075	50.0 (46.4, 53.5)*	3,953	37.4 (35.1, 39.8)	2,075	32.0 (29.2, 35.0)*	
Ontario	9,875	52.4 (50.4, 54.3)	7,182	44.0 (42.0, 46.0)*	9,875	38.5 (36.8, 40.1)	7,182	27.0 (25.4, 28.7)*	
Males	4,548	47.9 (45.3, 50.5)	3,216	42.7 (39.7, 45.6)*	4,548	35.3 (33.0, 37.7)	3,216	24.5 (22.1, 26.9)*	
Females	5,327	57.5 (54.8, 60.1)	3,966	45.4 (42.5, 48.3)*	5,327	41.5 (39.3, 43.7)	3,966	29.6 (27.2, 32.0)*	
Prairie Provinces	7,742	53.2 (51.3, 55.1)	7,731	44.5 (42.7, 46.3)*	7,742	37.1 (35.4, 38.8)	7,731	27.3 (25.8, 28.8)*	
Males	3,705	52.7 (49.9, 55.5)	3,582	44.0 (41.4, 46.6)*	3,705	36.3 (33.9, 38.7)	3,582	25.8 (23.8, 28.0)*	
Females	4,037	53.9 (51.2, 56.5)	4,149	45.0 (42.4, 47.6)*	4,037	37.9 (35.5, 40.3)	4,149	28.7 (26.7, 30.8)*	
British Columbia	4,706	53.4 (51.1, 55.8)	2,624	44.6 (41.5, 47.7)*	4,706	39.1 (36.9, 41.2)	2,624	26.7 (24.5, 29.1)*	
Males	2,203	48.5 (45.2, 51.9)	1,202	40.6 (36.5, 44.9)*	2,203	33.7 (30.7, 36.8)	1,202	25.7 (22.5, 29.3)*	
Females	2,503	59.1(55.6, 62.5)	1,422	49.0 (44.8, 53.3)*	2,503	44.4 (41.5, 47.3)	1,422	27.7 (24.6, 31.0)*	
65+ years									
Canada-wide	16,053	49.4 (45.9, 52.9)	18,945	37.8 (34.5, 41.2)*	16,053	29.4 (28.2, 30.6)	18,945	21.5 (20.5, 22.6)*	
Males	7,067	49.0 (44.5, 53.6)	8,254	37.8 (33.5, 42.2)*	7,067	27.3 (25.7, 29.0)	8,254	21.1 (19.6, 22.7)*	
Females	8,986	50.1 (44.2, 55.9)	10,691	37.9 (33.1, 42.9)*	8,986	31.2 (29.6, 32.9)	10,691	21.9 (20.5, 23.4)*	
Atlantic Canada	2,259	56.0 (48.2, 63.5)	4,113	45.8 (39.3, 52.5)*	2,259	32.2 (29.7, 34.9)	4,113	27.5 (25.5, 29.5)*	
Males	1,002	62.5 (51.5, 72.3)	1,788	46.0 (37.8, 54.5)*	1,002	32.9 (29.2, 36.9)	1,788	25.8 (22.9, 28.9)*	
Females	1,257	46.1 (35.1, 57.6)	2,325	45.4 (36.2, 55.0)	1,257	31.6 (28.5, 34.9)	2,325	29.0 (26.3, 31.8)	
Québec	3,582	45.6 (38.4, 53.1)	2,641	31.7 (24.9, 39.3)*	3,582	26.2 (24.2, 28.4)	2,641	19.6 (17.6, 21.7)*	
Males	1,613	53.3 (43.8, 62.5)	1,156	35.4 (26.3, 45.8)*	1,613	25.4 (22.4, 28.7)	1,156	20.6 (17.8, 23.8)*	
Females	1,969	36.5 (26.1, 48.4)	1,485	25.6 (17.3, 36.3)	1,969	27.0 (24.2, 30.0)	1,485	18.6 (15.9, 21.6)*	
Ontario	4,890	49.6 (43.2, 56.1)	5,430	39.3 (33.6, 45.2)*	4,890	28.4 (26.2, 30.7)	5,430	21.3 (19.4, 23.3)*	
Males	2,098	44.5 (36.4, 52.9)	2,347	38.3 (30.7, 46.5)	2,098	25.4 (22.4, 28.7)	2,347	21.6 (18.8, 24.8)	
Females	2,792	57.9 (47.0, 68.1)	3,083	40.4 (32.3, 49.0)*	2,792	31.0 (27.7, 34.4)	3,083	20.9 (18.5, 23.6)*	
Prairie Provinces	3,064	47.6 (41.8, 53.4)	4,669	40.7 (34.3, 47.5)*	3,064	30.3 (27.7, 33.0)	4,669	23.7 (21.5, 26.1)*	
Males	1,341	46.2 (38.5, 54.1)	2,040	34.6 (27.8, 42.2)*	1,341	27.8 (24.0, 32.1)	2,040	20.7 (17.9, 23.8)*	
Females	1,723	50.2 (41.2, 59.2)	2,629	49.8 (39.0, 60.7)	1,723	32.5 (29.2, 36.0)	2,629	26.4 (23.2, 29.8)*	
British Columbia	2,258	54.2 (46.3, 61.8)	2,092	34.7 (27.2, 43.1)*	2,258	35.3 (32.4, 38.2)	2,092	20.2 (17.8, 22.9)*	
Males	1,013	53.9 (43.5, 64.0)	923	39.5 (28.9, 51.1)	1,013	32.0 (27.9, 36.5)	923	18.6 (15.1, 22.6)*	
Females	1,245	54.5 (43.1, 65.6)	1,169	28.8 (19.6, 40.2)*	1,245	38.2 (34.4, 42.1)	1,169	21.7 (18.4, 25.3)*	
Notes.

Data are presented as mean (95% CI).

* Significantly different from 2018, p <0.05.

Figure 2 Proportion of Canadians accumulating ≤ 2 h per day of recreational screen use on work/school days across ages and regions.

Among adults aged 18–64 years, the proportion of individuals accumulating ≤2 hours/day of recreational screen use on workdays and non-workdays decreased for all groups (all p < 0.05). Among adults 65+ years, work-day decreases were seen among males in Atlantic Canada, Québec, the Prairies, and among females in Ontario and BC. On non-workdays, decreases were seen among all groups except females in Atlantic Canada and males in Ontario. (all p < 0.05).

Discussion

The purpose of this article was to describe age- and gender-specific regional trends in MVPA and screen time for Canadians aged 12+ years from 2018 to 2021. Our results suggest substantial regional and age-related variations in how MVPA levels were impacted by the COVID-19 pandemic across Canada. For recreational screen time, although the magnitude of changes varied across regions, a significant reduction in the proportion of individuals accumulating ≤2 h per day was seen in the majority of regions and age groups. The present study extends the previous reports of Colley & Watt (2022) and Colley and Saunders (Colley & Saunders, 2023a; Colley & Saunders, 2023b), all of which examined changes in MVPA and recreational screen time at the national level only.

When comparing all forms of MVPA, several trends become apparent. Among youth, Atlantic Canada and BC did not see a reduction in overall MVPA or active transportation levels in the fall of 2020 or 2021. Atlantic Canada was the only region which did not see a reduction in school-related MVPA in either 2020 or 2021, while BC was the only region which did not see a change in recreational MVPA levels when both genders were combined. There are a variety of potential explanations for the resilience of MVPA levels on the East and West coasts, including policy approaches (e.g., the “Atlantic Bubble” which limited the spread of COVID-19 early in the pandemic and allowed for less restrictions Patil, 2020; Bignami, 2021) and a relatively warm climate in BC which may make active transportation more feasible than other provinces. As noted earlier, Atlantic Canada and BC also experienced relatively lower levels of community transmission when compared to other parts of the country, which allowed for a greater level of “normalcy” in Atlantic Canada in particular, when compared to Québec, Ontario, and the Prairies in 2020. Limiting initial community spread in future pandemics may be one of the best ways to avoid disruptions to activity levels for residents within any given region.

Another trend identified in the data was that Ontario, Canada’s largest province, stood out as following a trend consistently worse than other regions. Ontario was the only province where the proportion of youth meeting MVPA recommendations and the amount of daily recreational MVPA remained depressed in the fall of 2021, when compared to 2018. Ontario was also one of two regions which did not observe an increase in adults aged 65+ years meeting the MVPA recommendation in 2021. As noted above, these trends may have been due to a number of factors, including rates of community transmission, the number and duration of restrictions, and the clarity of public health messaging. In particular, the low recreational MVPA, school-related MVPA and active transportation observed in Ontario among youth in 2021 may be partially due to relatively higher level of stringency of restrictions in Ontario at this time, when compared to other regions (Cameron-Blake et al., 2021). These restrictions included closing schools and both indoor and outdoor recreational facilities in April of 2021 (Canadian Institute for Health Information, 2023), greatly limiting opportunities for activity among youth. Future research should further investigate the relationship between specific policy approaches in each region and movement behaviours, to identify which decisions may have helped to support MVPA in Atlantic Canada and BC in 2020 and 2021.

The trajectory of MVPA from 2018 to 2021 differed markedly across age groups. While the proportion of youth meeting the MVPA recommendation decreased by 12 percentage points in the fall of 2020 and by 6 percentage points in 2021, it was relatively stable throughout the pandemic among adults aged 18–64 years and increased by 3 percentage points in the fall of 2021 among adults aged 65+ years. This is a remarkable difference that highlights the importance of considering the way that public health policies may have inequitable impacts on different age groups in future pandemics. Although there were clear differences in the trajectories of males and females at the national level, these were not as apparent at the regional level. Nationally, activity levels of female youth remained depressed in 2021, when compared to 2018, which was not observed in males of the same age. However, males and females in this age group followed similar trajectories in every individual region for both overall and recreational MVPA. This may be due to lack of sample size at the regional level, when compared to the national sample. It is also worth highlighting that while there were few obvious gender-related differences in the trajectories themselves, there remain clear differences in the total amount of MVPA and sedentary behaviour accumulated by males and females, respectively.

Although MVPA was more resilient in some regions and age groups than others, with very limited exceptions the sustained increases in recreational screen time were seen in all regions and age groups. This is concerning, given the links between high levels of recreational screen time and both physical and mental health (Carson et al., 2016; Ekelund et al., 2016; Saunders et al., 2020). These findings highlight the need for policy approaches to help limit the impacts of recreational screen time, as well as research to investigate the intended and unintended consequences of any such policies.

Strengths and limitations

The current study has several strengths and limitations. The CCHS is a large annual health survey, representative at national and sub-national levels, which has used consistent questions to ascertain MVPA and sedentary behaviour for Canadians aged 12+ years since 2015. To our knowledge this is the first such study to examine trajectories of MVPA and screen time across regions of Canada before and after the onset of the COVID-19 pandemic. This study was limited by its reliance on self-reported MVPA and recreational screen time, both of which have are known to have potential sources of error and bias (Prince et al., 2008; Prince et al., 2020; Adamo et al., 2009). Due to sample size constraints at the regional level, we were not able to examine whether our findings differed across sociodemographic groups, different cultural groups, or levels of physical ability. The questionnaires used in this study did not ask about the types or timing of recreational screen time, both of which may influence the health impacts of screen use (Saunders et al., 2022a). Further, the psychometric properties of the questions used to assess MVPA and screen time in the CCHS have not been evaluated. Finally, we did not assess school- or work-related screen time, which likely increased during the pandemic as many schools and offices transitioned to schooling and working from home and may have different impacts on health and well-being (Kuzik et al., 2022; Saunders et al., 2022b).

Conclusions

Our findings suggest that the MVPA levels of Canadian youth dropped significantly after the onset of the pandemic, and these changes varied greatly across regions. Although some regional variation remained, activity levels of adults aged 18–64 were relatively stable in the fall of 2020 and 2021 and increased for adults aged 65+ years. Recreational screen use increased across all regions and age groups with very limited exceptions. These results highlight the differences and similarities in activity and screen time trajectories across the Canadian population and suggest the need for additional research to identify best practices for promoting healthy movement behaviours during future pandemics.

Additional Information and Declarations

Competing Interests

Author Contributions

Human Ethics

Data Availability

Travis Saunders has received received personal fees for leading a report on the measurement of sedentary behaviour for the Public Health Agency of Canada (PHAC), and honoraria for presenting on the health impact of physical activity and sedentary behaviour to school groups. Rachel Colley declares no conflicts of interest. The views expressed in this article are solely those of the authors and do not reflect those of Statistics Canada.

Travis Saunders conceived and designed the experiments, prepared figures and/or tables, authored or reviewed drafts of the article, and approved the final draft.

Rachel C. Colley conceived and designed the experiments, performed the experiments, analyzed the data, prepared figures and/or tables, authored or reviewed drafts of the article, and approved the final draft.

The following information was supplied relating to ethical approvals (i.e., approving body and any reference numbers):

The Canadian Community Health Survey falls under the Statistics Act, and therefore does not require institutional ethics approval.

The following information was supplied regarding data availability:

The data from the Canadian Community Health Survey are available for researchers meeting Statistics Canada criteria through Research Data Centres and through the Statistics Canada website.

Information on how to order CCHS data is available here: https://www150.statcan.gc.ca/n1/en/catalogue/82M0013X.

The process required to access data through a Research Data Centre is available here: http://www.statcan.gc.ca/eng/rdc/process.

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
