# Peer review of "Regional trends in the moderate-to-vigorous intensity physical activity and screen time of Canadians before and during the COVID-19 pandemic"

_PeerJ, doi:10.7717/peerj.16913_

## Round 0.1 · original submission · Major Revisions

This paper is interesting and will add new information about physical activities during the pandemic with respect to regional differences in Canada, but before we can consider it to be published as the reviewers have clearly stated, the article needs major revision of the study design elements, presentation of tables and graphs, and some additional data analyses and rewriting in several places.

Please make those changes item for item for each reviewer and resubmit so that we can consider another round of review on time and arrive at a timely decision.

·

Basic reporting

The english is easy to read however as a non-Canadian I have trouble to put your results in perspective and therefore the manuscript becomes less interesting than it could be.
Your aim was to describe differences in life style habits based on where the participants live, this because of possible health consequences and future pandemics. But you don’t write how Canada was affected by the virus nor what kind of restrictions that were applied. Was public transportation open? Schools? Work places? Gyms? Was there a difference between the regions? Elderly seemed to increase their PA, and then mostly transportation wise, is it possible to put your numbers into perspective?
For instance, in most regions boys were active 20 min less and girls 10 min less during 2020, but not in BC. These boys did not follow the pattern of PA during 2020 compared to other regions, why is that? If you aim to present data according to region I would like an explanation why this is important or how the regions differ.

Experimental design

I miss the exact wording for the questions asked in the manuscript. Do you ask about intensity for PA? The same thing holds for screen time, sometimes you refer to recreational screen time but in the method section you only write that the question asked the participants to estimate average screen time. Not recreational.

Validity of the findings

According to how you present the questions in the method section as of today, you may not say that people adhered to the PA-guidelines. According to what is written there was no specification that the PA should be of moderate to vigorous intensity. Other researchers have suggested that low intensity PA increased during covid-19 but that does not necessarily mean that the population studied got healthier.

-You miss to inform the reader of the number of participants for each answer in the tables, likewise I miss information about the response rate for each year as well as a background analysis. How do your respondents differ from the population. You say that you weighed the results, how was that done and from what baseline? Are your numbers representative? You present a lot of p-values, without presenting the n it is difficult to understand whether that is of importance, for instance a 1% difference may be significant if you have 50000 respondents.

Additional comments

Thank you for letting me review this manuscript.
Overall, the scope is interesting and I believe that it is important to learn from what happened during the Covid-19 pandemic. In addition, the large amount of data that you present makes comparison between the different regions within one country even more interesting. Still there are some major shortcommings that needsto be adressed before publication.

Smaller comments:
-In the abstract, reading the first to paragraphs it is not obvious that adults are included it says 12+, which might refer to youths. In addition, reading the result it is not clear whether you mean PA or screen time.
-Table 2, total physical activity, please describe what this means in more detail.
-Was the mean age the same for responding youths for all years reported?
-When were restrictions removed, was the 2021-response really after covid-19? Could one assume that life had returned to “normal”?
-Please specify what screen time is, computers, cell phones, Tv? And did you precise “recreational”?
-There seems to be sex-differences, why do you only present these in the supplemental files without much of a discussion? If your aim is to describe how life style habits changed based on age and region isn’t sex another aspect that is important from a public health perspective?
-You may remove some of the references from unpublished work, it doesn’t add anything.
-All participants provided written informed consent, how about children, did their legal guardian consent?
-269-270 Was it overly warm during covid? Otherwise, wouldn’t commuting in BC be the same for all years?
-The paragraph on social media in the discussion is somewhat of topic since you don’t mention social media anywhere else in the manuscript. It may play a role but then you also need to write something about that you assume that a substantial part of youth screen time is social media-based.


To sum up, I believe that this manuscript has potential but needs elaborate work to reach its full potential.
Best regards

·

Basic reporting

• Some recommendations for strengthening text:
o Please update to “moderate-to-vigorous intensity physical activity” throughout the manuscript (e.g., Lines 63,65).
o Please ensure you are including the comparison group when presenting findings (e.g., Line 164: larger decrease compared to…).
o Ensure acronyms are provided at first use, then use consistently throughout (e.g., Lines 70-71, Line 167, etc.).
• I encourage authors to provide stronger rationale for this work:
o This study examines the important topic of physical activity (PA) and screen time (ST) among Canadians prior to and during the COVID-19 pandemic. The work by Colley & Saunders that is currently under review seems to be very similar to the presented work, with the main difference being examination of regional differences of PA and ST. Given this, I encourage the authors to strengthen their rationale in the background for why this work is important, and what this would add to the literature. The authors address this to some degree in lines 82-85, but there is a need to build upon this (i.e., authors state there are likely regional differences – why is this important to know?). This rationale should also be in line with study methodology, which currently does not statistically examine differences between regions.
• Literature references
o The Colley & Saunders reference (The ongoing impact of the COVID-19 pandemic on physical activity and screen time among Canadian adults) appears twice in the reference list.
o Line 92: Please provide a citation for CCHS.

Experimental design

• I suggest the authors increase the specificity of their research question to better reflect the work that was done. For example, consider mentioning sex and age, as these seem to be a big focus of the manuscript.
• I have some methodological concerns:
o I suggest reporting the psychometric properties of the measures used for PA and ST, if available.
o For the PA items:
 Please include the specific items that were used to measure PA, including for each domain. Were these open response? (i.e., Were respondents asked to provide time in minutes per day?)
 How was PA intensity determined?
 Why were values greater than 2 hours per day of any domain flagged as outliers? What basis was there for this cut-off? Was this done for both youth and adults? Given this, it could be misleading to present minutes per day of PA, since it is not an accurate reflection of what was reported.
 Please confirm in the methods whether the PA domain items were the same for youth and adults.
o For the ST items:
 Please include the specific items that were used to measure ST.
 It appears from both the methods and the discussion that the CCHS items did not allow for distinguishing between recreational and non-recreational screen time. If this is the case, take care not to refer to ST within the manuscript as recreational or to youth as meeting/not meeting ST guidelines, etc.
o Please provide more information about how the weights were developed (Lines 131, 149).

Validity of the findings

• I have some suggestions for improving the presentation of findings.
o Throughout the results section, the authors should take care to distinguish between “statistically/not statistically different” (e.g., line 206).
o Please include a demographic table for the sample included in the study.
o It was difficult to process all the data across the 7 tables and 2 figures (plus the supplemental tables). There are so much data presented, much of which are not fully addressed in the discussion section. I suggest that the authors streamline the results, which would also allow for a more robust discussion of the implication of these findings. Some examples are included below. (Also, please see my comment above regarding presentation of min/day of PA and the 2 hour cut-off).
 As mentioned above, be sure the study purpose accurately reflects what was done. It appears from the results that comparison by sex and age were an important part of the study goal. If so, consider presenting the findings in this way (similar to what as done in the supplement). I expand on this thought below.
 I strongly recommend presenting the data as figures – specifically line graphs. This would make it easier for readers to examine trends over time (which seem to be the main research goal). For example, you could present data by comparison group (age and/or sex) for both PA and ST over time as line graphs.
• If the authors decide to keep the tables, consider presenting data from either Table 1 or 2 (not both), including n’s and having “Mean (95% CI)” as a header rather than a footnote.
 Further, I think the domain-specific PA is very interesting and authors should allow for more ease in comparing the trends across domains.
• Some suggestions for the discussion section:
o Take care in the discussion section when comparing regions, sex, and age, since (to my understanding) statistical comparison was only across time.
o As mentioned above, the authors should consider which findings they want to focus on for their paper, and then expand on these more within the discussion. Currently, most of the discussion seems to be a re-statement of the results, and there is a need to expand on 1) how these findings compare to the larger literature base and 2) the implication of these findings. For example, in lines 274-279, the findings are re-stated and the discussion seems to be limited to: “As noted above, this is likely the result of multiple factors and is worthy of additional research”. Similar comments for the next paragraph (lines 281-287).

Additional comments

• The title suggests that PA and ST will be compared by region, but this is not in line with study methodology, which compares PA and ST trends over time within regions.

---

## Round 0.2 · accepted · Accept

The reviewers have recommended acceptance of this publication. Congratulations!

·

Basic reporting

Best Travis Saunders and Rachel Colley,
I am delighted to read your revised work. You have really made an effort in answering the comments I made as well as the comments of the other reviewer. I find the manuscript greatly enhanced and was happy to read the section about the different Canadian regions which made a great difference in understanding the importance of your work.
I have two minor comments that you may or may not address, either way I find your manuscript suitable for publication.
-Table 2 and 8, you write “data are presented as mean” shouldn´t it be percentages?
-Do you mean gender or sex?

Experimental design

I have no further comments on the experimental design after you have revised the manuscript according to earlier comments.

Validity of the findings

I have no further comments.

Additional comments

Good luck with publication and thank you for letting me review your work!

·

Basic reporting

The authors have adequately addressed my concerns.

Experimental design

The authors have adequately addressed my concerns.

Validity of the findings

The authors have adequately addressed my concerns.

Additional comments

The authors have adequately addressed my concerns.